# CryoEM structures of membrane pore and prepore complex reveal cytolytic mechanism of Pneumolysin

**Katharina van Pee, Alexander Neuhaus, Edoardo D'Imprima, Deryck J Mills, Werner Kühlbrandt*, Özkan Yildiz***

Department of Structural Biology, Max Planck Institute of Biophysics, Frankfurt am Main, Germany

**Abstract** Many pathogenic bacteria produce pore-forming toxins to attack and kill human cells. We have determined the 4.5 Å structure of the ~2.2 MDa pore complex of pneumolysin, the main virulence factor of *Streptococcus pneumoniae*, by cryoEM. The pneumolysin pore is a 400 Å ring of 42 membrane-inserted monomers. Domain 3 of the soluble toxin refolds into two ~85 Å $\beta$-hairpins that traverse the lipid bilayer and assemble into a 168-strand $\beta$-barrel. The pore complex is stabilized by salt bridges between $\beta$-hairpins of adjacent subunits and an internal $\alpha$-barrel. The apolar outer barrel surface with large sidechains is immersed in the lipid bilayer, while the inner barrel surface is highly charged. Comparison of the cryoEM pore complex to the prepore structure obtained by electron cryo-tomography and the x-ray structure of the soluble form reveals the detailed mechanisms by which the toxin monomers insert into the lipid bilayer to perforate the target membrane.

**\*For correspondence:** werner. kuehlbrandt@biophys.mpg.de (WK); Oezkan.Yildiz@biophys. mpg.de (ÖY)

## Introduction

Gram-positive bacteria, including the common wound-infecting *Staphylococcus aureus*, the virulent food-borne pathogen *Listeria monocytogenes*, and *Streptococcus pneumoniae* that causes pneumonia, utilize cholesterol-dependent cytolysins (CDCs) to attack and kill mammalian and human cells (*Gilbert, 2005*; *Tweten, 2005*). The bacteria produce and release CDCs as water-soluble monomers that attach to cholesterol-containing cell membranes, where they assemble into large, 200–500 Å cytolytic pores (*Olofsson et al., 1993*). X-ray structures of pneumolysin (PLY) from *S. pneumoniae* (*Lawrence et al., 2015*; *Marshall et al., 2015*; *van Pee et al., 2016*) show that the soluble toxin monomers are roughly rod-shaped and consist of four domains (D1-D4). Sequence comparison suggests that all CDCs have the same domain structure and therefore insert into target membranes in essentially the same way (*Tweten et al., 2001*). Although numerous biochemical studies have addressed the arrangement of monomers in the CDC prepore or pore complex (*Hotze and Tweten, 2012*; *Tilley et al., 2005*) and how they might penetrate the lipid bilayer (*Shatursky et al., 1999*), the detailed structure of the pore complex, and hence the exact mechanism of membrane insertion is unknown. The size heterogeneity of CDC prepores and pores has so far precluded structure determination at high resolution. We determined the structure of the ring-shaped ~2.2 MDa PLY pore complex by cryoEM. Together with our 2.4 Å x-ray structure of soluble PLY (*van Pee et al., 2016*) and a map of the prepore complex obtained by electron cryo-tomography, we can now describe the entire process of membrane attachment, prepore and pore formation in near-atomic detail. Given the high degree of sequence conservation amongst CDCs, it is likely that the same mechanism of membrane insertion holds for related bacterial toxins.

## Results

### CryoEM structure of the PLY pore complex

Ring-shaped pore complexes of wildtype PLY (PLY$_{WT}$) forming on unilamellar, cholesterol-containing liposomes were solubilized with detergent. The size and homogeneity of solubilized PLY pores is detergent-dependent, as indicated by negative-stain and cryoEM (*Figure 1—figure supplement 1*). Pore diameters varied from 310 to 500 Å in DDM or 350 to 400 Å in Cymal-6. Replacing the detergent Cymal-6 by amphipol A8-35 (*Althoff et al., 2011*; *Lu et al., 2014*) resulted in a stable population that was sufficiently homogenous for single-particle cryoEM. 2D class averages indicated a roughly even distribution of views (*Figure 1—figure supplement 2A*). Pore complexes were rings of 42 subunits with an aggregate molecular mass of 2.2 MDa. In total, 6461 ring images were combined to generate a 3D map with 42-fold symmetry at 4.5 Å resolution (*Figure 1*; *Figure 1—figure supplements 2B* and *3*,). The final map of the pore complex has an outer diameter of 400 Å and a total height of 110 Å, of which ~80 Å protrudes from the membrane surface (*Figure 1*).

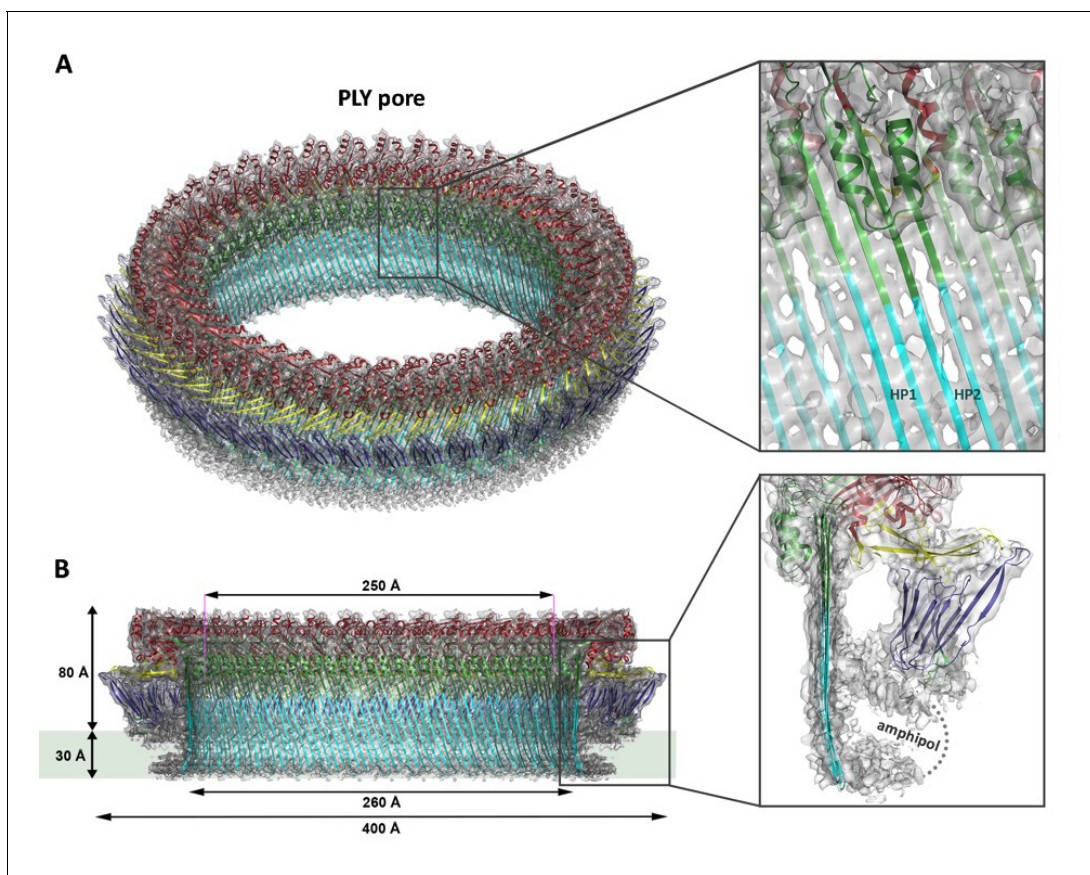

**Figure 1.** Overall structure of the PLY pore complex. (**A**) Single-particle cryoEM map of PLY at 4.5 Å resolution with fitted model. The four PLY domains (D1–D4) are red (D1), yellow (D2), green/cyan (D3) and blue (D4). Inset: refolded β-hairpins (HP1 and HP2) fitted to the map. Cyan β-strands have refolded from helix bundles in D3 of the soluble form. (**B**) Cross-section with overall dimensions of the pore complex. The grey bar indicates the position of the lipid bilayer. Inset: side view of membrane-inserted monomer with toroid density of disordered amphipol (broken line).

The following figure supplements are available for figure 1:

**Figure supplement 1.** Negative stain and cryoEM of PLY solubilized in DDM, Cymal-6 and Amphipol.

**Figure supplement 2.** Image processing of PLY rings.

**Figure supplement 3.** Local resolution estimate of the PLY monomers and of the complete pore complex.

Where possible, individual domains of the PLY x-ray structure (*van Pee et al., 2016*) were moved manually into the 4.5 Å cryoEM map as rigid bodies. Rebuilding of domains that refolded upon membrane insertion and readjustment of secondary structure elements and sidechains within domains yielded an atomic model of the pore complex (*Figure 1*). Comparison to the soluble form (*Figure 2A,B*) revealed a complete reorganization of the toxin upon membrane insertion. Of the four protein domains D1-D4 in the x-ray structure, D1 and D4 fitted the map of the pore complex with minimal modifications (*Figure 2—figure supplements 1–3*, *Video 1*). In the membrane-inserted form, the loop linking D1 to D2 refolds into a helix (α3a) at the interface between the rearranged domains D1, D2 and D3 (*Figures 2A,E*, *3* and *4*; *Figure 2—figure supplement 4*). In D4, the highly conserved undecapeptide loop (*Figure 3*) that renders PLY cholesterol-specific (*Soltani et al., 2007b*) was shifted by up to ~9 Å into the cryoEM map (*Figure 2—figure supplements 2* and *3*, *Video 1*). The undecapeptide loops of adjacent protomers in the pore complex are located in one plane on the outer membrane surface of the target cell, where they interact closely with the lipid head groups (*Figures 1B* and *2A,C*). In the linear rows of soluble PLY monomers that are found in the PLY crystal structures (*Lawrence et al., 2015*; *Marshall et al., 2015*; *van Pee et al., 2016*), the distance between the loops of neighbouring monomers is ~14 Å. In the pore complex this distance decreases to 4–5 Å, which enables an interaction of the loop that connects β-strands 18 and 19 (β18/19) in D4 with the β22/23 loop in D4 of the next monomer along the ring (*Figure 2—figure supplements 2* and *3*, *Video 1*). The close proximity of loop β18/19 that contains Asp403, Thr405, and His407 of one monomer to loop β18/19 and the uncedapeptide containing Trp433 of the adjacent monomer suggests a critical role of these D4 loops not only in receptor recognition, but also in oligomer formation.

Domains 2 and 3 undergo massive rearrangement and refolding (*Figure 2A,B*) in the membrane-inserted form. The elongated two-stranded β-sheet of D2 rotates from its vertical position in the x-ray structure by 90° around a short glycine linker to an orientation parallel to the membrane plane in the pore complex. D2 connects D1 to D4, and its rotation results in a roughly linear translation of D1 by 35 Å towards the membrane surface and by 30 Å towards the pore centre. Otherwise, D2 required only minor adjustments of secondary structure for an optimal fit to the cryoEM map (*Figures 1B* and *2A*, *Figure 2—figure supplement 5*).

By contrast, the structure of D3 changes entirely upon membrane insertion. In the soluble toxin, the central, five-stranded β-sheet in D3 is flanked by two bundles of short α-helices (*Figure 2B*). In the pore complex, both bundles refold into four 85 Å β-strands, which form two parallel β-hairpins that insert into and traverse the lipid bilayer (*Figure 2A*). The β-hairpins of neighbouring subunits coalesce into one extensive, 168-strand β-barrel with an inner diameter of 260 Å (*Figure 1B*). The new β-strands can be traced unambiguously, because they are straight, apparently rigid and continuous with four β-strands in the D3 x-ray structure that are preserved in the membrane form. The two new, long β-hairpins are inclined by 20° relative to the membrane normal, in good agreement with predictions for perfringolysin O on the basis of cysteine crosslinking experiments (*Sato et al., 2013*). The chain trace in the refolded domain is confirmed by the positions of bulky densities for large side-chains in the β-barrel (*Figure 2D*; *Figure 2—figure supplement 4*). While five of the six helices in bundles HB1 and HB2 refold into β-strands in the pore complex, one remains intact. Conversely, the loop and fifth β-strand in the central β-sheet of D3 refold into a helix (*Figure 2A,B*). Together, the two α-helices form a helix-turn-helix (HTH) motif (*Figure 2—figure supplement 4*). In the ring, the HTH motifs of the 42 subunits line up in one plane on the exoplasmic side of the molecule and form an α-barrel inside the β-barrel (*Figures 1B* and *4*), restricting the pore diameter locally to 250 Å (*Figure 1B*).

The outside of the membrane-inserted part of the β-barrel is almost entirely hydrophobic (*Figure 5A,E*) and covered by the toroid density of disordered amphipol that replaces the membrane lipid (*Figure 1B*), as in the cryoEM structures of amphipol-solubilized membrane proteins (*Althoff et al., 2011*; *Liao et al., 2013*). The inside pore surface is highly polar, with three aspartates and two glutamates forming a 15 by 9 Å patch of negative charge, flanked above and below by positive charges (*Figure 5B,F,G*). Apart from the hydrogen bonds between the 168 β-strands in the barrel and the interactions of helix α3a with D1 and β-hairpin one (*Figure 2E*, *Figure 2—figure supplement 4*), three other factors contribute to pore stability: (1) the surface and charge complementarity of the membrane-inserted monomers (*Figure 5C*); (2) the alternating positive and negative charges of helices α13a and α13b in the α-barrel (*Figure 5D*); and (3) ionic interactions between

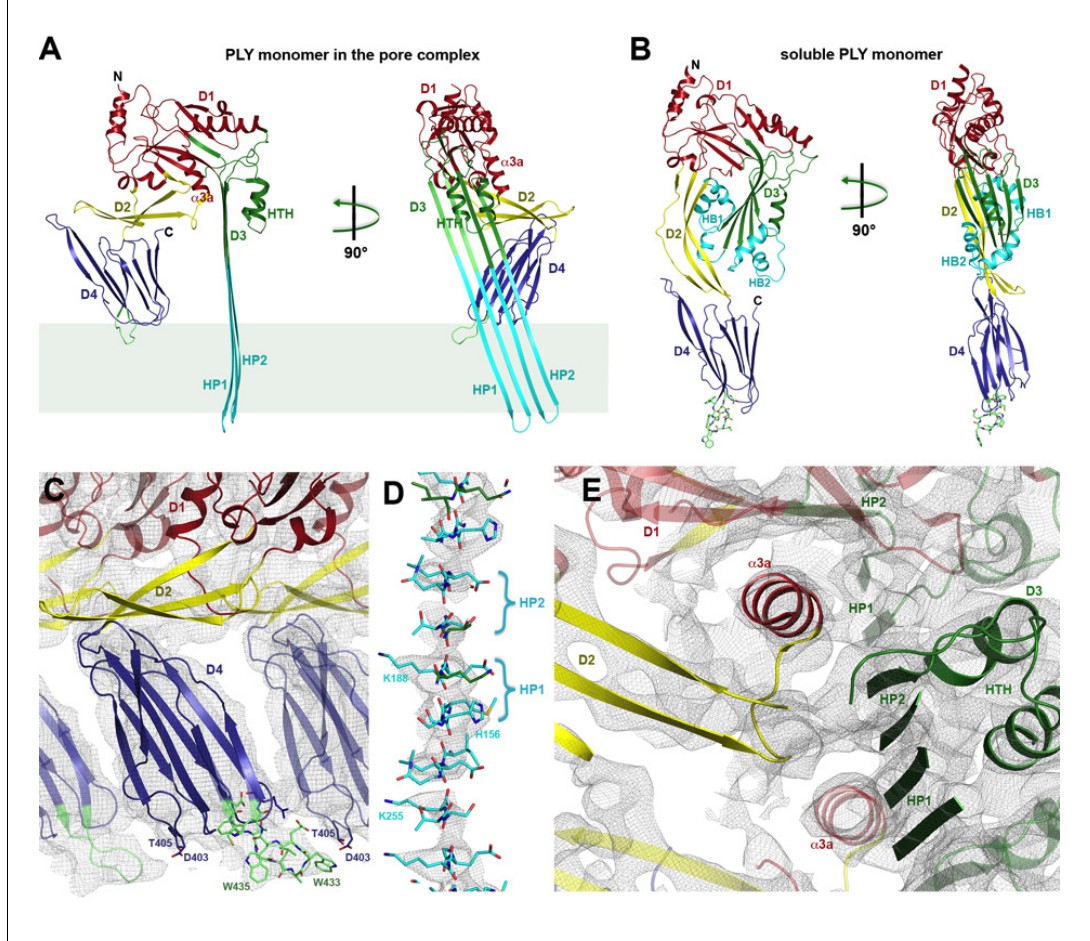

**Figure 2.** Soluble and membrane-inserted PLY monomer. One subunit of membrane-inserted form of PLY in the cryoEM structure (**A**) and x-ray structure of soluble PLY (*van Pee et al., 2016*) (**B**) seen from the side (left) and from the pore center (right). Both helix bundles (HB1 and HB2, cyan) in the PLY monomer refold to form two long, membrane-spanning β-hairpins (HP1 and HP2). The upper end of the β-hairpins is sandwiched between a helix-turn-helix motif (HTH, green) on the inside and helix α3a (red) of the next monomer on the outside of the pore. One of the helices is a refolded β-strand of D3 (green). Helix α3a is the refolded linker that connects D2 to D1 in the soluble monomer. (**C**) D4 contains the conserved undecapetide (green sticks) that confers cholesterol specificity to PLY. The peptide includes three resolved Trp sidechains, one of which interacts with Thr405 in the adjacent monomer. (**D**) Cross section through a segment of the 168-strand β−barrel with resolved bulky sidechains. The two β-hairpins of one monomer are labelled. (**E**) Helix α3a interacts with HP1 and D1 of the neighbouring monomer, stabilizing the pore complex.

The following figure supplements are available for figure 2:

**Figure supplement 1.** Stereo view of domain 1 presented as ribbon-and-stick model.

**Figure supplement 2.** Superposition of domain 4 of the PLY monomer in the crystal structure (blue) and in the pore complex (green), viewed from the pore center.

**Figure supplement 3.** Stereo view of domain 4 in the pore complex with potential inter- and intramolecular interactions of loops viewed from two sides.

**Figure supplement 4.** Stereo view of domain 3 with the upper part of the β-barrel and the new helix-turn-helix motif (HTH) that forms the α-barrel inside the pore.

**Figure supplement 5.** Stereo view of domain 2 seen along the ring axis.

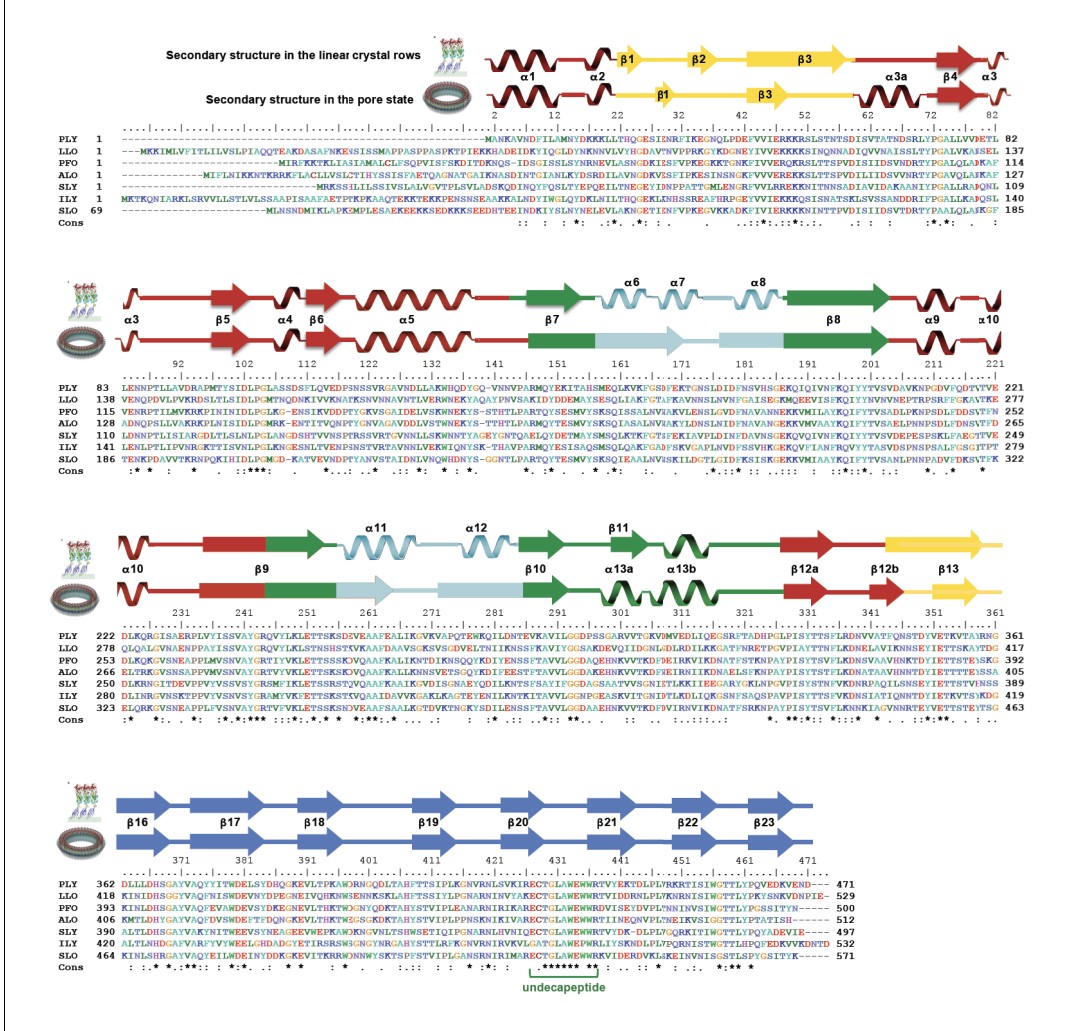

**Figure 3.** Sequence alignment of selected cholesterol-dependent cytolysins. Elements of secondary structure are shown for the crystal structure of soluble PLY (above) and for the cyroEM structure of the pore complex (below). Colors of the secondary structure elements and residue numbers correspond to the PLY crystal structure (*Figure 2B*; pdb code 5aod). Asterisks indicate conserved residues.

charged sidechains in adjacent β-strands of the pore barrel (*Figure 5G*). In particular, Asp168 and Glu170 in β-strand β7 are in a good position for forming a salt bridge with Lys271 in β-strand β10 of the next-door monomer (*Figure 5G*).

## Three stages of PLY pore formation

To investigate the structures of the PLY prepore and pore complex in native membranes prior to detergent solubilisation, we incubated cholesterol-containing liposomes with purified PLY and generated 3D volumes by electron cryo-tomography (cryoET). The liposomes were studded with numerous ring-shaped complexes of 290 to 360 Å outer diameter (*Figure 6A*). Assemblies were identified as PLY pores or prepores by the absence or presence of a lipid bilayer within the ring (*Figure 7A*). Pore and prepore complexes were classified independently and processed by subtomogram averaging (*Figure 6B*). The subtomogram average map of the prepore complex at a resolution of 22 Å (*Figure 6C*) had an outer diameter of 320 Å and accommodated 34 subunits (*Figure 6D*). Of the available PLY crystal structures (*Lawrence et al., 2015*; *Marshall et al., 2015*; *van Pee et al., 2016*), the model 5cr6 (*Marshall et al., 2015*) fitted the map best (*Figure 6—figure supplement 1*), indicating that in this structure the soluble toxin is in the pre-pore state. The subtomogram average map of the pore complex at 27 Å resolution (*Figure 6C*) indicated a slightly larger diameter of

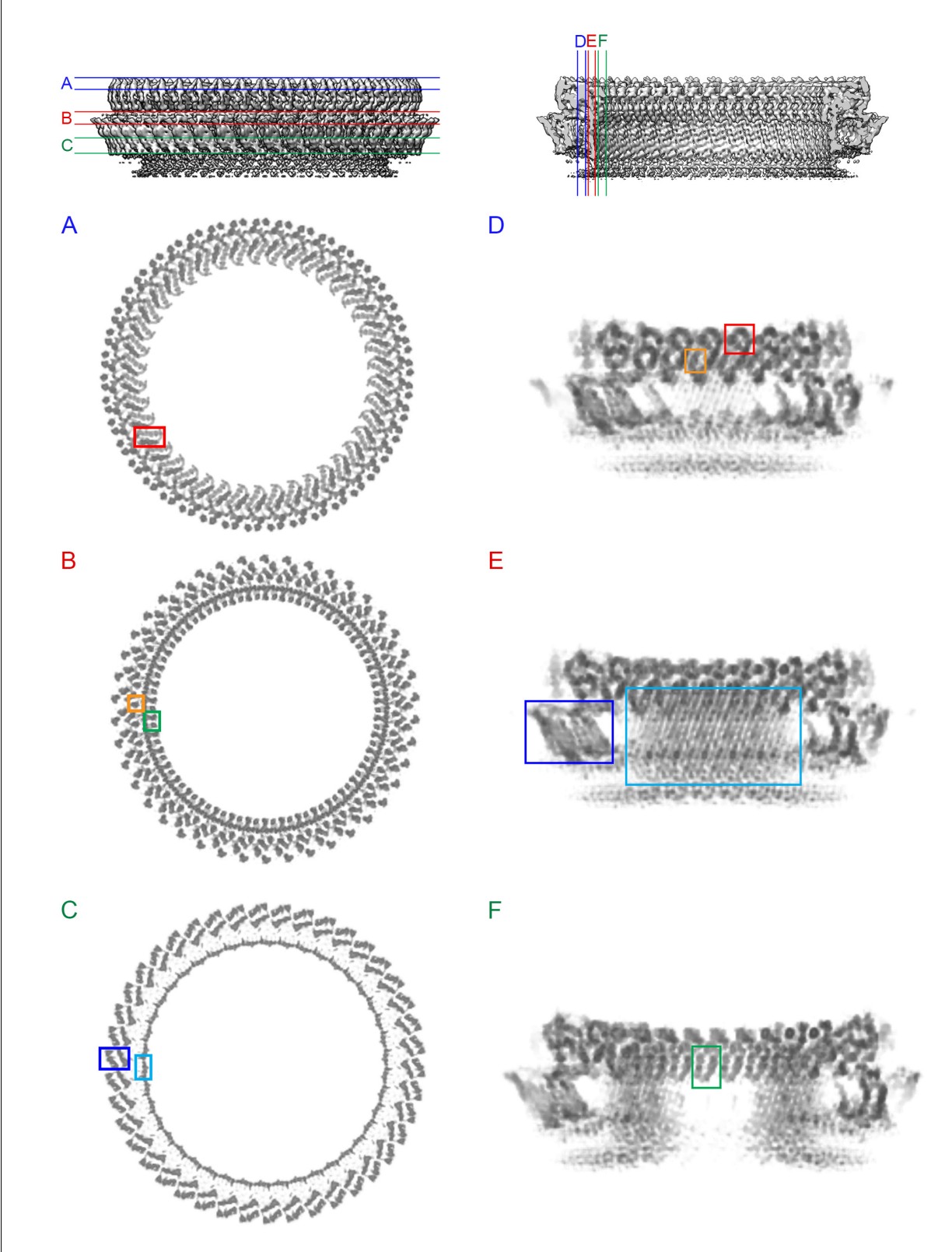

**Figure 4.** Intermolecular interactions in the 4.5 Å cryoEM map of the PLY pore complex. Slices parallel (**A**–**C**) and perpendicular (**D**–**F**) to the membrane reveal interactions of secondary structure elements between adjacent monomers in the pore complex. Section planes are indicated on top. Note that the scale of sections D to F is 50% larger to show molecular detail more clearly. The β5-α4-β6 region in domain D1 and the long membrane-parallel helix α5 (red box in **A** and **D**) alternate around the top of the pore complex. On the outside, the 168-strand β-barrel is flanked by helix α3a (orange box

*Figure 4 continued on next page*

*Figure 4 continued*

in **B** and **D**) and by the α-barrel of the helix-turn-helix motifs (green box in **B** and **F**) on the inside. The 4-strand β-sheets of domain D4 (blue box in **C** and **E**) are offset against D4 of the next monomer, forming a ring of 8-strand β-sheets. The 85 Å-long β-strands forming the 168-strand β-barrel are clearly resolved (cyan box in **C** and **E**).

around 350 Å, but likewise consisted of 34 subunits (*Figure 6E*). The cross-section profile resembled that of the high-resolution cryoEM map of the pore complex closely (*Figure 1*), indicating that solubilisation with detergent or amphipol does not change the structure of the membrane form significantly. The model of membrane-inserted PLY fitted the subtomogram average of the pore complex without significant adjustment, including the 20° inclination of the long β-hairpins relative to the membrane normal.

Atomic force microscopy (AFM) had revealed two different forms of PLY prepores in planar bilayers, rising 110 or 80 Å above the membrane surface (*van Pee et al., 2016*). The low form has the same height as the pore complex, but the lipid bilayer is still intact. We propose that in this lower prepore, D1, D2 and D4 have moved to their positions in the pore assembly, while D3 has not yet refolded and the long, membrane-spanning β-hairpins have not yet formed. We assign the tall form to an early prepore state, which then rearranges into the lower, late prepore state. The cross-section profile (*Figure 6B,D*) shows that the early prepore is essentially a ring-shaped side-by-side arrangement of soluble PLY monomers (*Figure 6—figure supplement 1*), similar to that in the crystal lattice (*Lawrence et al., 2015*; *Marshall et al., 2015*; *van Pee et al., 2016*). While the late prepore was observed for 13% of the rings on planar bilayers examined by AFM, it was not observed in tomographic volumes of PLY on liposomes, implying that it is an intermediate, transient state which inserts more readily into curved liposomes than into planar bilayers. Cross-sections of pore complexes in liposomes (*Figure 6B,E*) show that the lipid bilayer around the prepores and pores is curved, whereas in the AFM images it was flat (*van Pee et al., 2016*). Prepore stability and membrane insertion thus seem to be related to membrane curvature, such that pores form preferentially in lipid bilayers with a convex curvature.

## Determinants of lytic activity

To understand the functional role of individual sidechains in oligomerization, prepore and pore formation, we mutated Asp168 that, as the structure suggests, might be involved in forming a salt bridge between adjacent subunits to alanine (PLY$_{D168A}$). We also deleted Ala146 and Arg147 (PLY$_{\Delta146/147}$) in the loop that would clash with the last β-strand of the central β-sheet. The resulting mutants were characterized by negative-stain EM (*Figure 7A*), hemolytic activity assays (*Figure 7B*), and x-ray crystallography (*Figure 8*). X-ray data were collected to 2.45 and 2.5 Å resolution from crystals of PLY$_{D168A}$ and PLY$_{\Delta146/147}$ (*Table 1*). The structures were solved by molecular replacement with the x-ray structure of PLY$_{WT}$ (*van Pee et al., 2016*). As in PLY$_{WT}$, the mutant monomers crystallized in rows. Superposition of both structures on PLY$_{WT}$ showed no significant overall differences, although a detailed comparison indicated small changes in domain D2, helix bundle HB2 and in two of the D4 loops (*Figure 8*). Around the mutated residue in PLY$_{D168A}$ differences were restricted to sidechain orientations, while the deletion of two residues in PLY$_{\Delta146/147}$ caused significant conformational changes in the loop connecting helix α5 to strand β7 of the central β-sheet. This deletion also caused a slight shift of helix α5 towards D3 (*Figure 8*).

Even though the structures of PLY$_{D168A}$ and PLY$_{\Delta146/147}$ were very similar to PLY$_{WT}$, they displayed major differences in their membrane-binding and oligomerization behaviour. Like PLY$_{WT}$, PLY$_{D168A}$ bound to cholesterol-containing

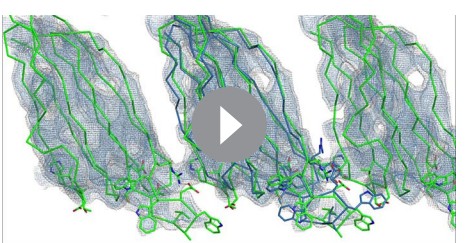

**Video 1.** Superposition of domain 4 of the PLY monomer from the crystal structure (blue) on domain 4 of the PLY monomer in the pore complex (green) rotated around an axis perpendicular to the membrane. Residues of the undecapeptide and selected residues in other loops are shown as sticks.

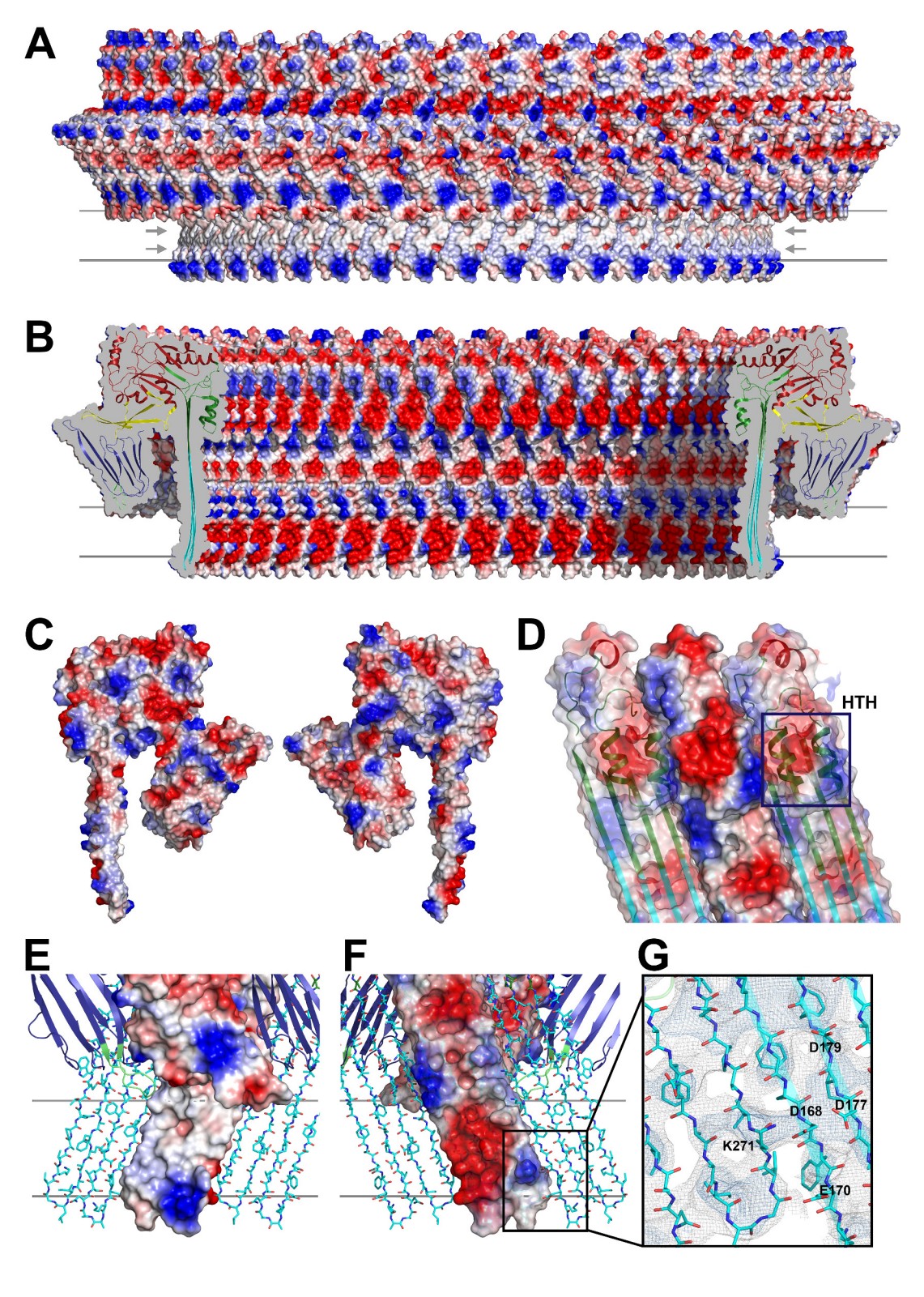

**Figure 5.** Charge distribution in the PLY pore complex. (**A**) Positive (blue) and negative charges (red) are evenly distributed on the polar outer surface of the pore complex. The membrane-inserted region is marked by a band of neutral hydrophobic residues (arrows). (**B**) The inner surface of the pore complex is highly charged. (**C**) Contact surfaces of adjacent PLY monomers in the pore complex. (**D**) Charge complementarity of the two helices in the helix-turn-helix motif forming the internal α-barrel. Positive charges are shown in blue and negative charges in red. (**E**) The membrane-inserted region

*Figure 5 continued on next page*

*Figure 5 continued*

of the β-barrel is hydrophobic on the outside and negatively charged on the inside (F). The inset (G) shows map density for the salt bridge between Asp168 of one PLY monomer with Lys271 in the next monomer along the ring. Glutamates and aspartates forming the negatively charged patches on the inner surface of the β-barrel are drawn as sticks. The map is contoured at 5.0 and 6.0 σ.

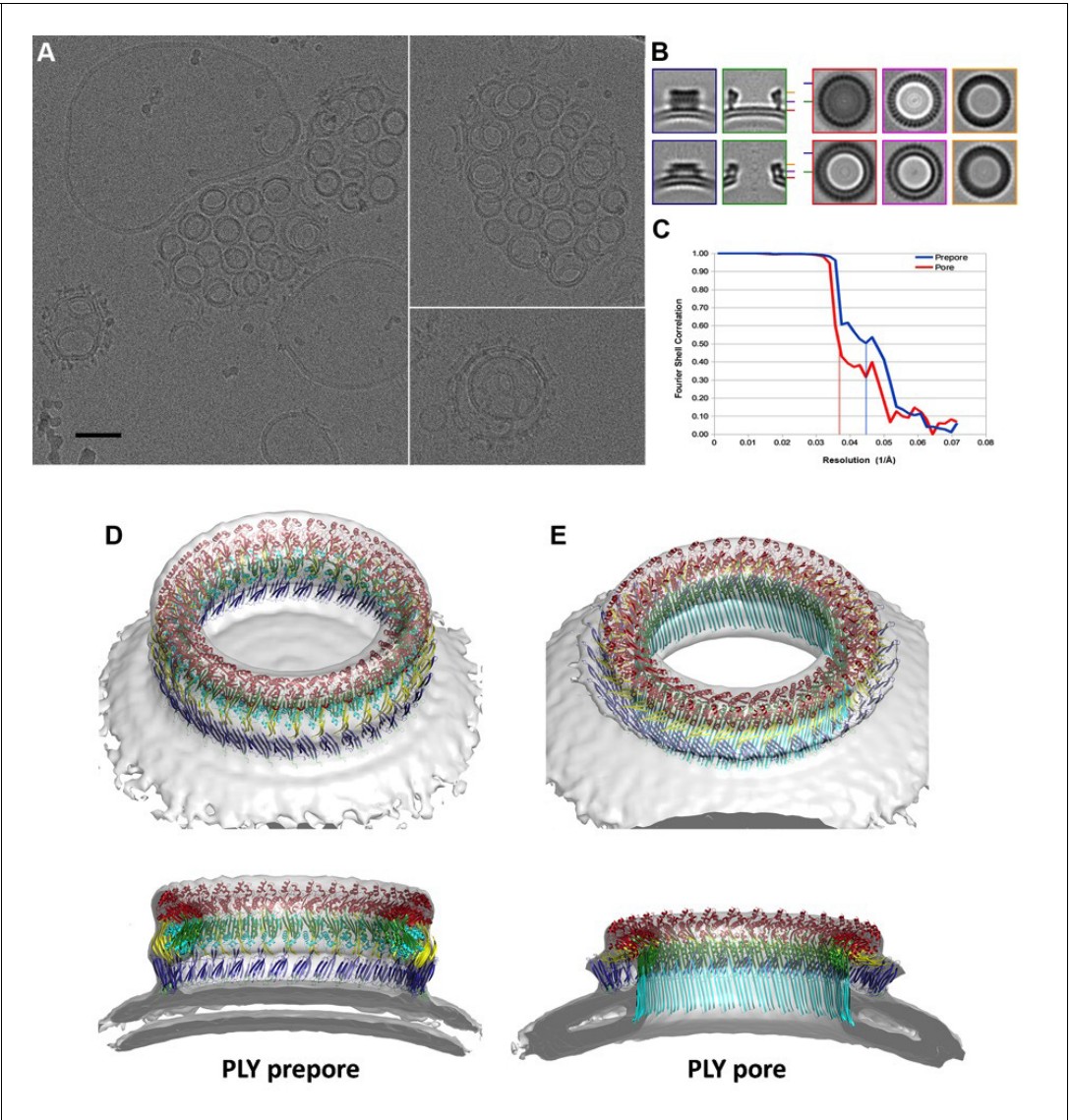

**Figure 6.** CryoET of PLY prepores and pores. (A) PLY assembles into prepores and pores upon incubation with cholesterol-containing liposomes. Scale bar, 50 nm. (B) Sections through subtomogram average volumes of the prepore (top) and pore complex (below). Left panel: sections perpendicular to the membrane; right panel: sections parallel to the membrane. Colors indicate section planes. (C) Fourier shell correlation for subtomogram averages indicate 22 Å resolution for the prepore and 27 Å for the pore at $FSC_{0.5}$ or 20 Å and 21 Å resolution at at $FSC_{0.3}$. Oblique view (D) and cross section (E) of PLY prepore (left) and pore (right). Both maps accommodate 34 PLY monomers. The prepore map was fitted with the crystal structure of the soluble PLY monomer (*Marshall et al., 2015*), and the pore map with the cryoEM structure of the pore monomer (*Figure 2A*). PLY domains are red (D1), yellow (D2), green/cyan (D3) and blue (D4). The lipid bilayer is continuous in the prepore complex, but absent in the pore complex.

The following figure supplement is available for figure 6:

**Figure supplement 1.** CryoET map of the PLY prepore with rigid-body fitted x-ray structures of the water-soluble toxin that forms rows in the 3D crystals.

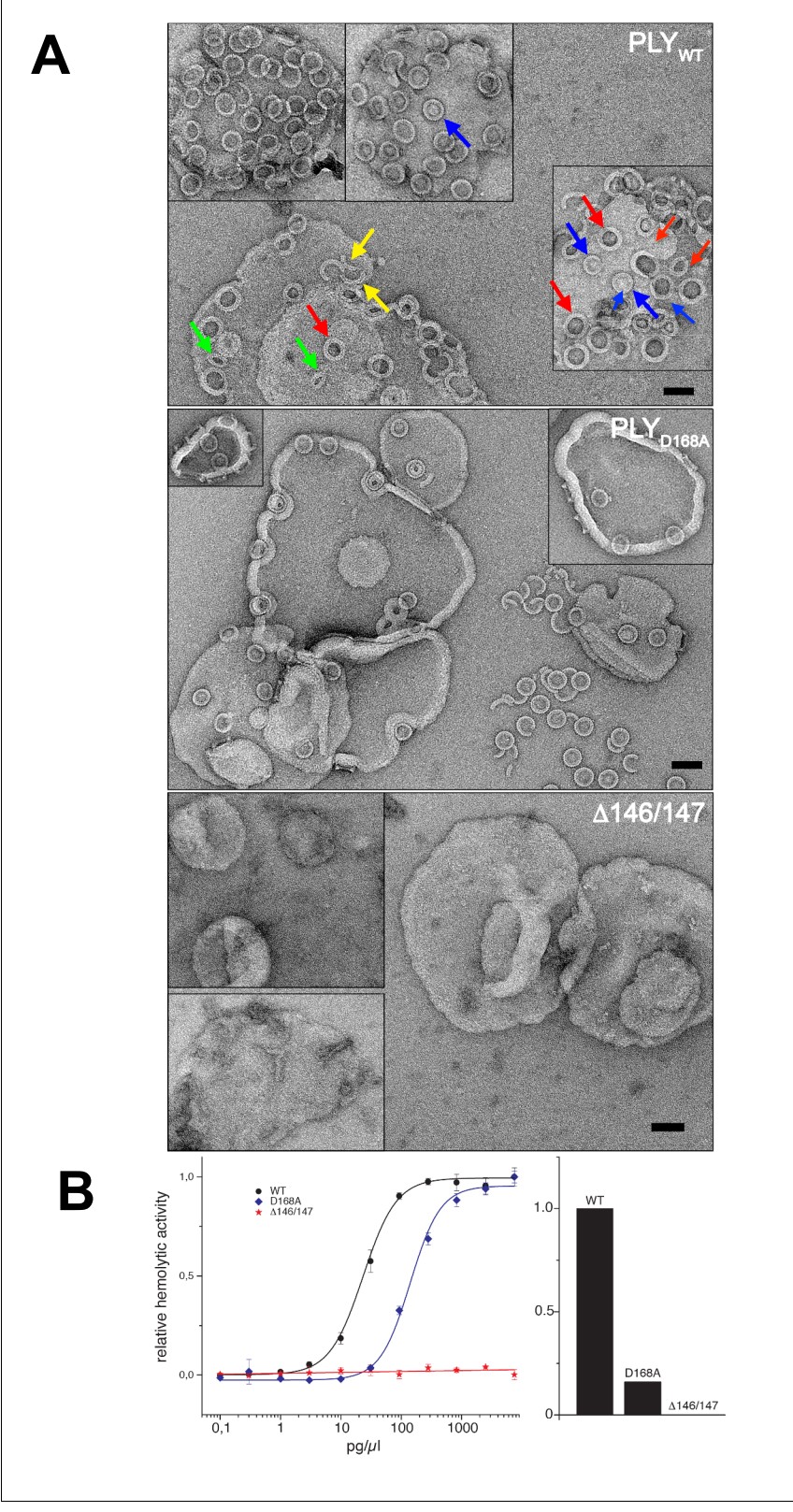

**Figure 7.** Membrane binding and hemolytic activity of PLY . (**A**) Wildtype PLY (PLY$_{WT}$) and PLY$_{D168A}$ form rings on cholesterol-containing liposomes. PLY$_{WT}$ lyses the majority of liposomes, while PLY$_{D168A}$leaves them mostly intact. Lipid-filled rings with a narrow rim (blue arrows) are prepores, while rings with a wider rim that do not contain lipid (red arrows) are pores. Slits (green arrows) and arcs (yellow arrows) are observed occasionally, but mostly PLY forms complete rings. Mutant PLY$_{D168A}$ prepores detach easily from the liposomes due to reduced binding affinity, and then break into fragments.
*Figure 7 continued on next page*

*Figure 7 continued*

Curves indicate the hemolytic activity of PLY$_{WT}$, PLY$_{D168A}$, and PLY$_{\Delta146/147}$. PLY$_{\Delta146/147}$ is inactive, in line with the inability of this mutant to form oligomers on cholesterol-containing liposomes. Scale bar: 50 nm.

liposomes and oligomerized into rings (*Figure 7A*). However, unlike PLY$_{WT}$ that lysed the liposomes within a short time, liposomes incubated with PLY$_{D168A}$ remained intact for hours. Frequently, the rings detached from the liposomes, indicating they were prepores that had not yet inserted into the membrane. This is also reflected by the hemolytic activity, which was reduced by 80% compared to PLY$_{WT}$, highlighting the important role of Asp168 in ionic interactions between β-strands that stabilize the pore complex.

PLY$_{\Delta146/147}$ was not able to bind to the membranes at all and had no detectable hemolytic activity (*Figure 7B*). Ionic interactions between the β-strands thus contribute significantly to pore stability, and any disruption of these interactions compromises pore formation. Both mutations pinpoint

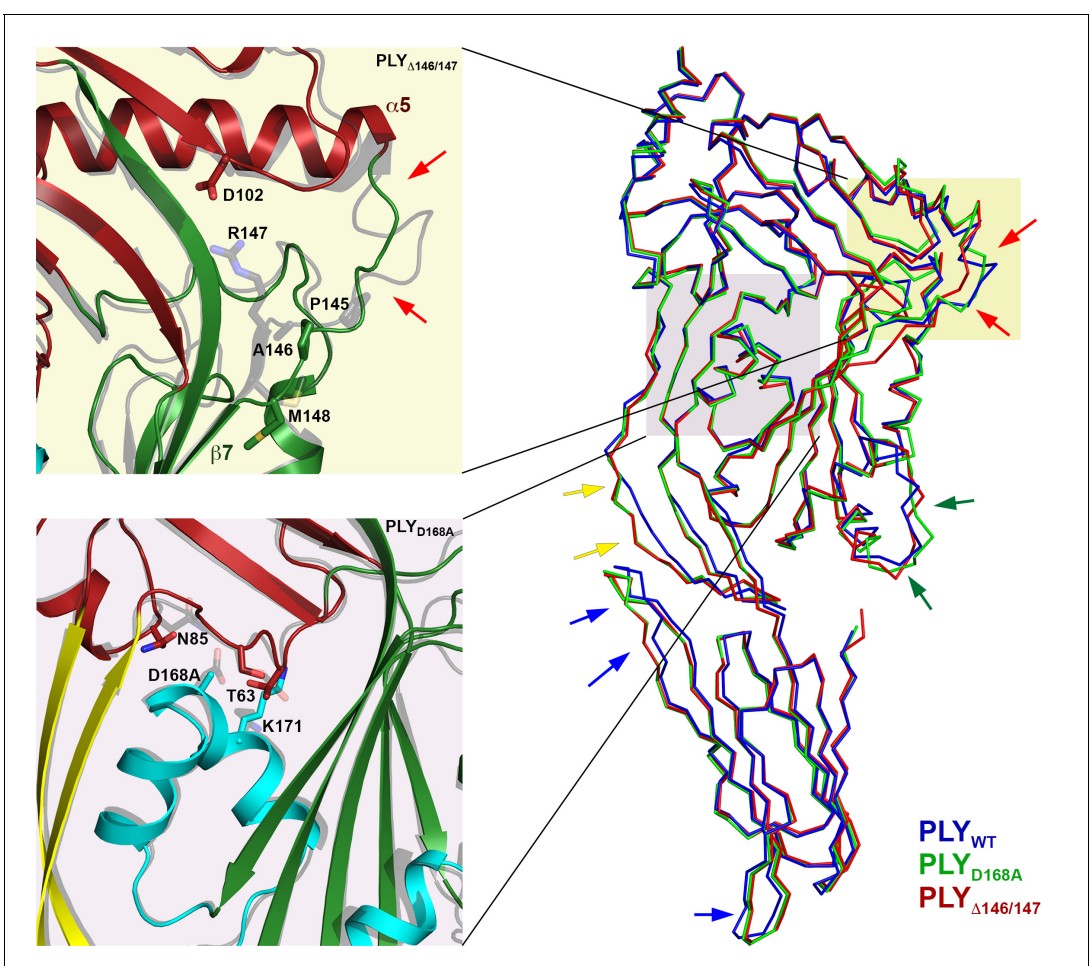

**Figure 8.** Location of functionally important PLY residues. (A) Ala146 and Arg147 in the loop that induces refolding of the last β-strand in the central D3 β-sheet into helix α13a in the late prepore and pore complex. Deletion of both residues renders the toxin inactive. (B) In the pore complex, Asp168 near the end of one long trans-membrane β-hairpin (HP1) forms a salt bridge with Lys271 in the other trans-membrane β-hairpin (HP2) of the adjacent monomer. Replacing Asp168 by alanine inhibits membrane insertion. (C) α-carbon traces in the x-ray structures of PLY$_{WT}$ (pdb 5aod), PLY$_{\Delta146/147}$ (pdb 5aof), and PLY$_{D168A}$ (pdb 5aoe). Minor differences between wildtype and mutant structures are visible in the loop regions of D4 (blue arrows); D2 (yellow arrows) and HB2 (green arrows). In PLY$_{\Delta146/147}$, one loop connecting D1 to D3 is also slightly different (red arrows).

**Table 1.** Data collection and refinement statistics.

| | PLY$_{D168A}$ (pdb-id 5aoe) | PLY$_{\Delta146/147}$ (pdb-id 5aof) |
|---|---|---|
| *Data collection* | | |
| Beamline | PXII @ Swiss Light Source x10sa | |
| Resolution (Å) | 40–2.5 (2.6–2.5) | 40–2.45 (2.55–2.45) |
| Wavelength (Å) | 0.9786 | 0.978 |
| Space group | $P2_1$ | $P\,2_1\,2_1\,2_1$ |
| Cell dimensions | | |
| a, b, c (Å) | 160.86 24.66 208.35 | 24.73 163.5 207.8 |
| α, β, γ (°) | 90, 90.26, 90 | 90, 90, 90 |
| Total reflections | 316531 (30235) | 137706 (13089) |
| Unique reflections | 59222 (5685) | 29722 (2937) |
| Multiplicity | 5.3 (5.3) | 4.6 (4.5) |
| Completeness (%) | 99 (100) | 91 (94) |
| Mean $I\,/\,\sigma I$ | 8.1 (1.4) | 9.1 (1.2) |
| Wilson B-factor | 46.75 | 45.05 |
| $R_{pim}$ | 0.08 (0.593) | 0.07 (0.607) |
| CC* | 0.998 (0.887) | 0.999 (0.792) |
| *Refinement* | | |
| Reflections used in refinement | 59207 (5683) | 29718 (2936) |
| Reflections in test set | 2962 (285) | 1486 (147) |
| $R_{work}/R_{free}$ (%) | 22.25/24.98 (32.9/33.7) | 20.87/23.44 (33.23/34.37) |
| CC(work)/CC(free) | 0.946/0.934 (0.769/0.737) | 0.955/0.930 (0.626/0.567) |
| Average B-Factor (Å$^2$) | 72.5 | 64.3 |
| No. atoms in AU | 7878 | 4008 |
| Protein | 7724 | 3856 |
| Water | 154 | 152 |
| r.m.s. deviations: | | |
| Bond lengths (Å) | 0.003 | 0.03 |
| Bond angles (°) | 0.72 | 0.69 |
| Ramachandran favored (%) | 96 | 96 |
| Ramachandran allowed (%) | 3.6 | 3.1 |
| Ramachandran outliers (%) | 0.5 | 0.8 |

Note: Values for the highest resolution shell are shown in parentheses

protein regions that are promising targets for drug development. Drugs that interfere with these interactions would render the toxin ineffective, and the bacteria that produce it non-pathogenic.

## Discussion

With the ongoing resolution revolution in cryoEM (*Kühlbrandt, 2014*), structures of macromolecular assemblies can now be determined at high resolution without crystals. This is proving especially useful for membrane proteins and membrane protein complexes (*Allegretti et al., 2015*; *Gu et al., 2016*; *Hahn et al., 2016*; *Letts et al., 2016*; *Liao et al., 2013*), which tend to be unstable and flexible, and often do not crystallize. In the case of CDC pore complexes, structure determination represents a particular challenge, because they consist of variable numbers of 30 to 50 monomers

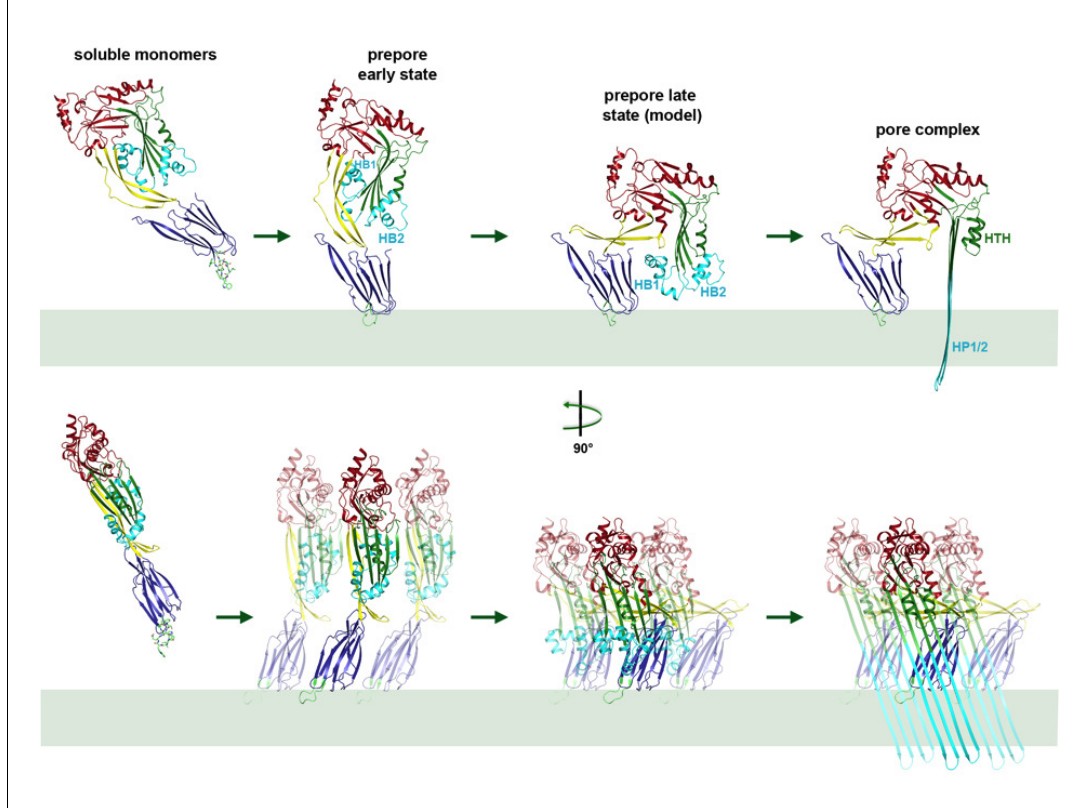

**Figure 9.** Mechanism of membrane insertion and pore formation. Stepwise conformational changes of PLY toxin during pore formation shown for one monomer (above, side view) and for three neighbouring monomers (below, view from the pore centre). In the first step, soluble PLY monomers attach to the surface of cholesterol-containing cell membranes via the conserved D4 undecapeptide (*Figure 2C*) to form circular oligomers of the early prepore. A 90° rotation of D2 moves D1 and D3 towards the membrane in the late prepore. Helix bundles HB1 and HB2 are poised above the membrane surface to refold into 85 Å β-hairpins HP1 and HP2. In the final step of pore formation, both hairpins traverse the hydrophobic membrane core and assemble into a 168-strand, 260 Å β-barrel. Reorganization of the PLY monomer exposes numerous charges on the inside of the β-barrel (*Figure 4*) that would destabilize the lipid bilayer and repel membrane lipids, resulting in pore opening and cell lysis.

(*Dang et al., 2005*; *Shatursky et al., 1999*; *Shepard et al., 1998*), and hence are intrinsically inhomogeneous. This fact has so far precluded structure determination of the membrane-attached or membrane-inserted form at high resolution. By a rigorous screen of the detergents used for solubilisation and purification of the pore complex from cholesterol-containing liposomes we were able to isolate a suitably homogenous population. Exchanging the detergent against amphipols appeared to stabilize rings of uniform size, as an important prerequisite for high-resolution single-particle cryoEM. The number of monomers in the amphipol-stabilized complexes was higher than in the prepores or the pores imaged by cryoET, even though the liposomes were prepared in the same way. This means that either the number of subunits in the ring varies from one liposome preparation to another, or that the rings rearrange into a more stable form upon detergent solubilisation, which we consider more likely. Interestingly, an earlier, low-resolution cryoEM structure of PLY pores and prepores in lipid bilayers deduced that there were 31 ± 3 subunits in the prepore and 38 or 44 in the pore complex (*Tilley et al., 2005*), in good agreement with our findings. Since the PLY pores in our subtomogram averages were smaller, consisting of 34 rather than 42 subunits, it seems that the rings can grow in the membrane by incorporating further subunits. Apparently, complexes of 42 monomers are more stable than both larger and smaller rings. The inherent variability in ring size restricts the number of particles that can be averaged in any one class. Moreover, the large ring-shaped assemblies are easily distorted and rarely, if ever, perfectly circular, which limits the accuracy to which they can be aligned by image processing. Both factors constrain the attainable resolution of the cryoEM reconstruction.

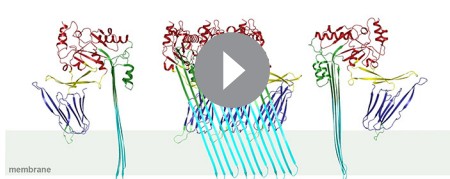

**Video 2.** Mechanism of pneumolysin pore formation. (1) Soluble PLY monomers attach to cholesterol-rich membranes by the cholesterol-binding undecapeptide (light green) of domain D4 (blue) and oligomerize into rings. For simplicity, only three ring subunits are shown. (2) Domain D2 (yellow) rotates by 90°, bringing domain D3 (green) with its two helix bundles (cyan) close to the membrane surface. (3) The helix bundles insert into the membrane and unfold into two trans-membrane 85 Å β-hairpins. β-hairpins of the 42 subunits in the ring merge into one large 168-strand β-barrel, which perforates the membrane.

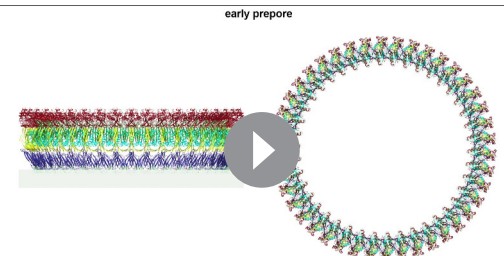

**Video 3.** Model of conformational changes in an entire ring of 42 PLY monomers from early prepore to late prepore to pore, seen from the side and from above.

Earlier cryoEM work, AFM and FRET studies have described the vertical height change of the pre-pore upon membrane insertion in terms of a collapse or unfolding of D2 (*Czajkowsky et al., 2004*; *Ramachandran et al., 2005*; *Tilley et al., 2005*). Our atomic model of the pore complex now shows that this change is due to a rigid-body rotation of D2, in which the structure of the domain remains intact. Time-lapse AFM of PLY and listeriolysin (LLO) on planar membranes indicated that the vertical height change is not directly linked to pore formation (*Mulvihill et al., 2015*; *van Pee et al., 2016*), in line with a mechanism that involves two different prepore states that we refer to as the early and late prepore. Our model of the late prepore monomer (*Figure 9*) is an intermediate between the x-ray structure of the soluble form and the cryoEM structure of the membrane-inserted form. The 5-stranded central β-sheet in the crystal structure of D3 fits into the cryoET map of the early prepore complex without any modification, but the late prepore model indicates a potential steric clash between the short β-strand in the central β-sheet of D3 and the adjacent monomer. The pivotal role of the loop that contains residues 145–147 connecting helix α5 of D1 to β7 of β-hairpin one is underlined by mutant PLY$_{\Delta 146/147}$ that was completely inactive (*Kirkham et al., 2006*) and did not even bind to liposomes (*Figure 7A*). The detachment of PLY$_{D168A}$ rings indicates that the mutant does not insert into the membrane as easily as PLY$_{WT}$ to form pores. We have shown by time-lapse AFM that pore formation is irreversible (*van Pee et al., 2016*). Therefore the rings that become detached from the membrane are prepores, not pores.

In earlier models of pore formation, the conserved domain D4 (*Figure 3*) was thought to be important for toxin targeting, receptor recognition and binding to cholesterol-containing membranes (*Farrand et al., 2010*; *Ramachandran et al., 2002*; *Soltani et al., 2007a*, *2007b*), whereas oligomer formation was mainly attributed to intermolecular interactions via D1 and D3 (*Lawrence et al., 2015*; *Ramachandran et al., 2002*). It has however been shown that D4 of streptolysin, pyolysin and LLO can oligomerise by itself on cholesterol crystals (*Harris et al., 2011*; *Weis and Palmer, 2001*) or erythrocyte ghosts (*Köster et al., 2014*). Our cryoEM structure confirms that D4 can indeed

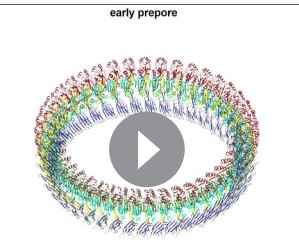

**Video 4.** Oblique view of PLY ring inserting into the membrane.

play a role in oligomer formation through intra- and intermolecular interactions of its loops (*Figure 2—figure supplements 2* and *3*). Loop $\beta$22/23 and the conserved undecapeptide loop with Trp433 at its tip appear to interact with loop $\beta$18/19 of the neighbouring monomer, thereby stabilizing the pore complex by inter- and intramolecular interactions (*Figure 2—figure supplements 2* and *3*).

The surface charge distribution on the toxin monomers is critical for CDC activity. Modification of surface charges affects ring formation, as we demonstrated by mutagenesis both for LLO (*Köster et al., 2014*) and PLY (*van Pee et al., 2016*). The alternating positive and negative charges of the HTH motif (*Figure 5D*) and the positive charges at the tips of the $\beta$-hairpins (*Figure 5E,F*) demonstrate the importance of charge complementarity for PLY ring formation. A similar HTH motif was found in the 8 Å cryoEM map of poly-C9 component of the human membrane attack complex (*Dudkina et al., 2016*). This HTH motif was already present in the soluble monomer of the C6 component (*Aleshin et al., 2012*) As in PLY, the HTH in the poly-C9 pore forms an α-barrel that determines the effective pore diameter.

Together with the x-ray structure of the soluble form, our cryoEM structures of the pore and pre-pore explain the step-by-step process of membrane insertion and pore formation in near-atomic detail. Movies showing the transition from the soluble monomer to the pore complex via the intermediate stages of the early and late prepore illustrate the complete mechanism of pore formation (*Videos 2–4*). In the early prepore, PLY monomers assemble side by side into rings on the membrane surface. In the late prepore, the D3 helices are poised for re-folding immediately above membrane surface. Helix unfolding may be a stochastic yet cooperative event, such that the spontaneous transition of one monomer triggers the refolding of its neighbours in the ring, comparable to a zipper. As the 168-strand $\beta$-barrel assembles in the membrane, the charged patches on its inside surface would repel any trapped hydrophobic lipid. Most likely the lipid is pushed out of the nascent pore by successively inserted toxin monomers, rather than being ejected in the form of micelles or small vesicles, as has been proposed for suilysin (*Leung et al., 2014*). Upon pore formation, the membrane potential collapses, the cytoplasm leaks out and the cell dies.

The mechanism of membrane insertion by rearrangement of the conserved domains 1–4 is likely to be the same for all CDCs. Therefore, compounds that interfere with refolding and membrane insertion of soluble monomers would prevent infection by *Streptococcus pneumonia* and other CDC-producing Gram-positive bacteria that attack human cells with similar pore-forming toxins. The 4.5 Å cryoEM structure of the PLY pore complex thus paves the way towards the design of new drugs that inhibit pore formation as a promising approach towards combatting infections by dangerous and wide-spread Gram-positive pathogens.

# Materials and methods

## Protein expression and purification

The gene coding for N-terminal His$_6$-tagged PLY was inserted into the pET15b vector. *E. coli* BL21 (DE3) cells transformed with the expression plasmid were grown in TB medium containing 50 μg · ml$^{-1}$ ampicillin. Protein expression was induced with 1 mM isopropyl-$\beta$-D-1-thiogalactopyranoside upon reaching an optical density of one at 600 nm. After 4 hr at 37°C the cells were pelleted, resuspended in Buffer A (50 mM Tris pH 7.0, 150 mM NaCl, 30 mM Imidazole, 5 mM $\beta$-mercaptoethanol) and disrupted with a Microfluidizer (M-110L, Microfluidics Corp., Newton, MA). PLY was purified on a HisTrap FF column, equilibrated with Buffer A. The protein was eluted in Buffer A containing 300 mM imidazole. Protein fractions were pooled and diluted in 50 mM Tris pH 7.0, 5 mM $\beta$−mercaptoethanol to a final NaCl concentration of 50 mM. The His$_6$-tag was removed by overnight cleavage with thrombin at 4°C. The protein was further purified on a HiTrap Q FF ion-exchange column equilibrated in Buffer B (50 mM Tris-HCl pH 7.0, 50 mM NaCl, 5 mM $\beta$-mercaptoethanol). PLY was eluted in Buffer B containing 170 mM NaCl. Protein fractions were pooled, concentrated to 10 mg · ml$^{-1}$ and stored at −80°C.

## Site-directed mutagenesis and hemolysis assays

Site-directed mutagenesis was performed with the QuikChange site-directed mutagenesis kit (Stratagene) according to the manufacturer's instructions with the wildtype construct as a template. All

constructs were verified by nucleotide sequencing. The hemolytic activity of PLY was determined by lysis of sheep red blood cells (SRBC) (*Darji et al., 1995*). Purified protein was serially diluted in hemolysis buffer (50 mM sodium phosphate pH 6.6, 150 mM NaCl, 5 mM DTT, 0.1% (v/v) BSA) in final volumes of 50 µl and incubated for 30 min at 37°C with 50 µl of a suspension of SRBC ($I0^8$ $ml^{-1}$). Release of hemoglobin was monitored by recording the absorbance at 405 nm. The amount of toxin necessary to lyse 50% of erythrocytes was determined and expressed as percentage of the value for $PLY_{WT}$. The absorbance upon incubation with 1% Triton-X100 was used as reference value for 100% lysis of erythrocytes. Three independent measurements were performed for each PLY mutant.

## Liposome preparation and amphipol stabilization

A lipid mixture containing 70 mol-% di-oleyl phosphatidyl choline (DOPC) and 30 mol-% cholesterol in chloroform was dried under a constant nitrogen stream. The lipid film was taken up in 50 mM Tris-HCl pH 7.0, 150 mM NaCl, 5 mM $\beta$-mercaptoethanol at a final concentration of 10 mg · $ml^{-1}$ and stirred overnight at room temperature. After three freeze-thaw cycles (liquid nitrogen, 37°C), the suspension was passed through an extruder to obtain unilamellar ~200 nm liposomes. The extruded liposomes were flash-frozen in liquid nitrogen and stored at −20°C. The liposome suspension was incubated with PLY at a final lipid-to-protein ratio of 1:2 (wt/wt) at 37°C for 30 min. For the preparation of PLY pore complexes, the proteoliposomes were solubilized at a final concentration of 0.56% Cymal-6 at room temperature overnight. Amphipol A8-35 was added in fivefold molar excess and the suspension was incubated for 30 min at room temperature. Detergent was removed by dialysis against 50 mM Tris-HCl pH 7.0, 150 mM NaCl, 5 mM $\beta$-mercaptoethanol at room temperature for 72 hr.

## Specimen preparation for negative-stain EM

PLY mutant pores were compared to wildtype protein formed on planar lipid layers or cholesterol-containing liposomes prepared as above. The liposome mixture was diluted to a final concentration of 0.5 mg · $ml^{-1}$ and 25 µg · $ml^{-1}$ PLY was added. Planar lipid bilayers and PLY were incubated in Teflon well plates. 25 µg · $ml^{-1}$ PLY in reaction buffer (150 mM NaCl, 50 mM Tris-HCl pH 7.0, 5 mM $\beta$-mercaptoethanol) was transferred into a well and overlaid with a droplet of a 0.5 mg · $ml^{-1}$ DOPC/cholesterol solution (1:1 molar ratio) in chloroform. Planar lipid layers formed upon chloroform evaporation. Wells were incubated for 30 min at 37°C, after which PLY-containing membranes were transferred to EM grids and stained with 1% (wt/vol) uranyl acetate. Negatively stained specimens were examined in an FEI Tecnai Spirit electron microscope at an acceleration voltage of 120 kV. Images were recorded on a 2 K Gatan CCD camera at a magnification of 25,000x–45,000x and ~1.0–1.5 µm underfocus.

## Single-particle cryoEM and image processing

Amphipol-solubilized PLY pores were diluted to a final concentration of 1 mg · $ml^{-1}$. A 3 µl aliquot was applied to freshly glow-discharged R2/2 holey carbon grids (Quantifoil Micro Tools, Jena, Germany), blotted for 3 to 4 s at 10°C and 100% humidity in a Vitrobot Mark IV (FEI). Preferential orientation of toxin rings on the air-water interface was overcome by carbon backing. Dose-fractionated 6.0 s movies of 30 frames with an electron dose of 1.02 e⁻ · ($Å^2$ · frame)$^{-1}$ were recorded after coma-free alignment (*Allegretti et al., 2014*) with an FEI Polara electron microscope operating at 300 kV with 0.5–3.6 µm underfocus and a specimen pixel size of 1.4 Å on a K2 Summit direct electron detection camera (Gatan, Pleasanton, USA) operating in counting mode with an energy filter slit of 20 eV. Movie frames were corrected for beam-induced motion with Motioncorr (*Li et al., 2013*) and again with UNBLUR (*Grant and Grigorieff, 2015*), which also applied a dose-dependent filter. The contrast transfer function of each image was determined using CTFFIND3 (*Mindell and Grigorieff, 2003*). A total of 12308 particle images were hand-picked from 983 micrographs in RELION 1.3 or 1.4 (*Scheres, 2012, 2015*) and extracted into 360-pixel boxes. Initial 2D classification indicated that rings with 42-fold symmetry were most common, although rings with different symmetries were also present. For 3D classification in RELION, the pore structure of suilysin (EMD-2983) low-pass filtered to 50 Å was used as a reference. For the next processing step, the unsymmetrized low-resolution PLY map was used as the reference. Subsequent 3D classification yielded three classes, of which two

had clear 42-fold symmetry. Rings that were distorted, damaged or of different size were sorted out at the 3D classification stage. A total of 6461 particles from 3D classes of 42-fold symmetry were auto-refined in RELION. A B-factor of $-175$ Å$^2$ for map sharpening was determined using the modulation transfer function of the K2 Summit detector. The resolution of the processed data after B-factor sharpening was 4.5 Å (FSC$_{0.143}$), with an estimated orientation accuracy of 0.75°. Local resolution was assessed with RELION 2.0 (*Kimanius et al., 2016*).

## CryoET of PLY prepores and pores

Preformed liposomes were incubated with PLY (1 mg · ml$^{-1}$) at room temperature for 30 min to obtain prepores, or at 37°C for 30–180 min to obtain pores. For cryoET the liposomes were diluted 1:3 with buffer (150 mM NaCl, 50 mM Tris-HCl pH 7.0, 5 mM $\beta$-mercaptoethanol) and then 1:1 with 6 nm gold particles conjugated with protein A (Aurion) as fiducial markers. Samples of 3 µl were applied to glow-discharged Quantifoil EM grids (R2/2, Cu 300 mesh), excess liquid was blotted off with filter paper (Whatman #4) and samples were vitrified by plunge-freezing into liquid ethane (*Dubochet et al., 1988*). Single-axis tilt series were typically collected from $-62°$ to $+62°$ at 2° increments and 3–4 µm underfocus with a total electron dose of 60–80 e$^-$ · Å$^{-2}$, on a Tecnai Polara electron microscope equipped with a field emisson gun operating at 300 kV (FEI, Hillsboro, OR), a post-column energy filter (GIF Quantum, Gatan) and a K2 summit direct electron detector (Gatan). Images were recorded in counting mode with a pixel size of 0.35 nm. Tilt series were CTF-corrected, binned $2 \times 2$ and aligned. Tomographic volumes were generated by weighted back projection in IMOD (*Kremer et al., 1996*).

## Sub-tomogram averaging

For particle picking and initial rounds of sub-tomogram averaging (*Frangakis and Hegerl, 2001*), tomograms were filtered by nonlinear anisotropic diffusion to enhance contrast. Ring-shaped complexes were picked manually in 3dmod. A total of 752 prepore complexes and 2400 pore complexes were picked for averaging. Ring volumes were pre-aligned in one plane in $2 \times 2$ binned unfiltered tomograms. All prepores had roughly the same size and shape and were averaged in PEET (*Nicastro et al., 2006*). The average volume was used as an initial reference for symmetry determination by sub-tomogram averaging with IMOD (*Kremer et al., 1996*). Symmetries of 42-fold or lower were applied to each prepore volume on a trial-and-error basis. Volumes were aligned and averaged in an iterative process, which converged within about 15 iterations on an average volume with clear 34-fold symmetry. The best 70% of the prepore volumes contributed to the final average, in which 34 PLY monomers were resolved in the ring.

Pore complexes were separated into seven classes according to their shape and diameter by principal component analysis and clusterPCA in IMOD (*Kremer et al., 1996*). The best class with 844 pore complexes was used for further processing as described above. In the final pore complex average, individual subunits were clearly distinguished in one third of the ring, indicating some heterogeneity of ring size and symmetry (*Figure 5B*). The number of PLY monomers in the pore was determined as 34 from the angular distance between distinct subunits in the average volume and the ring diameter. As for the prepores, all averaging steps were performed in PEET (*Nicastro et al., 2006*). The resolution at FSC$_{0.5}$ was 27 Å for the pore and 22 Å for the prepore (*Figure 6c*). At FSC$_{0.3}$, the resolution of the pore and prepore complex would be 21 Å and 20 Å, respectively.

## Model building and analysis

Model building was performed in COOT (*Emsley and Cowtan, 2004*) based on the x-ray structure of the water-soluble PLY monomer (pdb 5a0d; (*van Pee et al., 2016*). Initially, individual domains in the x-ray structure of soluble PLY were fitted manually into the cryoEM map as rigid bodies using COOT. Refolded or flexible protein regions were re-fitted manually, followed by geometry regularization in COOT. The complete backbone of PLY was traced in the pore complex map. Densities of bulky side chains were observed in well-ordered regions. The transmembrane $\beta$-hairpins, which form upon unfolding of HB1 and HB2 in domain 3 were readily fitted to the map density. The pore model with 42 subunits was generated in UCSF Chimera (*Pettersen et al., 2004*). Sub-tomogram average maps of the PLY prepore were fitted in UCSF Chimera with the x-ray structures of the PLY monomer (pdb 5cr6; [*Marshall et al., 2015*]). Sub-tomogram averages of the pore complex were fitted with

the cryoEM structure of the membrane-inserted form. Figures were drawn with PyMol (*Schrö-dinger, 2015*). FSC curves of model versus map were calculated using the EMAN package (*Tang et al., 2007*).

### Crystallization, data collection, structure determination, and refinement

Intitial crystallization trials were carried out with a protein concentration of 8 mg · ml$^{-1}$ (PLY$_{D168A}$) or 10 mg · ml$^{-1}$ (PLY$_{\Delta 146/147}$) after addition of glycerol to a final concentration of 10% in 96-well plates by vapour diffusion. 300 nl of protein solution were mixed with 300 nl of a commercially available crystallization solution (PGA Screen from Molecular Dimensions, JB Screen Classics I from Jena Bioscience, and Classics I Suite from Qiagen) in a pipetting robot (Mosquito, TTP Labtech). Hanging drops were incubated over 100 µl of reservoir solution at 18°C. Initial crystals hits were refined by varying the protein-to-reservoir ratio with a drop volume of 3 µl, incubated over 500 µl reservoirs at 18°C in 24-well plates. PLY$_{D168A}$ crystals grew after one day at 18°C in a 3 µl drop of 1.5 µl protein solution (8 mg · ml$^{-1}$ in 10% glycerol) and 1.5 µl of reservoir solution (0.1 M sodium cacodylate pH 6.5, 1% PGA-LM, 0.2 M potassium bromide and 0.2 M potassium thiocyanate). PLY$_{\Delta 146/147}$ crystals grew in a few days at 18°C in a 3 µl drop of 2.0 µl protein solution (10 mg · ml$^{-1}$ in 10% glycerol) and 1.0 µl of reservoir solution (0.1 M imidazole pH 6.5 and 1.2 M sodium acetate). Crystals were transferred to reservoir solution as a cryo-protectant and flash-frozen in liquid nitrogen. Data were collected at beamline PXII (Paul Scherrer Institute, Villigen, Switzerland) under a constant stream of cold nitrogen gas (100 K). Data processing, integration and scaling was performed with the XDS package (*Kabsch, 1993*). Structures of the PLY mutants were solved by molecular replacement with PLY$_{WT}$ (pdb-id 4AOD) as a search model using PHASER (*McCoy, 2007*) in the CCP4 software package (*Collaborative Computational Project, Number 4, 1994*). The initial electron density map was improved by cycles of density modification, automatic model building in RESOLVE (*Terwilliger, 2004*) and refinement by REFMAC (*Murshudov et al., 1997*). The model was subjected to iterative rounds of rebuilding into 2F$_o$-F$_c$ and F$_o$-F$_c$ maps using COOT (*Emsley and Cowtan, 2004*) and refined with the phenix.refine subroutine in the PHENIX program suite (*Zwart et al., 2008*). Data collection, refinement, and model statistics are summarized in *Table 1*. Figures were generated with PyMOL (*Schrödinger, 2015*).

### Accession numbers

The cryoEM density map has been deposited in the Electron Microscopy Data Bank under the accession number EMD-4118. The structure coordinates have been deposited in the protein data bank under accession number 5LY6 (PLY pore complex), 5AOE (PLY$_{D168A}$) and 5AOF (PLY$_{\Delta 146/147}$)

## Acknowledgements

We thank Janet Vonck for discussion and advice on image processing, Sabine Häder and Heidi Betz for technical assistance, Sebastian Schaupp for help with screening solubilisation conditions and negative-stain EM, and Juan Castillo for computer support. This work was funded by the Max Planck Society.

## Additional information

#### Competing interests

WK: Reviewing editor, *eLife*. The other authors declare that no competing interests exist.

#### Funding

| Funder | Grant reference number | Author |
| --- | --- | --- |
| Max-Planck-Gesellschaft | DepartmentSB | Katharina van Pee<br>Alexander Neuhaus<br>Edoardo D'Imprima<br>Deryck J Mills<br>Werner Kühlbrandt<br>Özkan Yildiz |

The funders had no role in study design, data collection and interpretation, or the decision to submit the work for publication.

### Author contributions

KvP, Data curation, Investigation, Methodology, Writing—original draft, Performed biochemical experiments, Carried out single-particle cryoEM and processing, Interpreted the data and wrote the paper; AN, Investigation, Methodology, Performed cryo-tomography and sub-tomogram averaging and analysis of data; ED, Investigation, Methodology, Interpreted the data and wrote the paper; DJM, Investigation, Methodology, Devised the cryoEM data collection strategy; WK, Resources, Formal analysis, Funding acquisition, Writing—review and editing, Interpreted the data and wrote the paper; ÖY, Conceptualization, Data curation, Formal analysis, Supervision, Validation, Investigation, Visualization, Methodology, Writing—original draft, Project administration, Writing—review and editing, Initiated and directed the project, Interpreted the data and wrote the paper

### Author ORCIDs

Werner Kühlbrandt, http://orcid.org/0000-0002-2013-4810
Özkan Yildiz, http://orcid.org/0000-0003-3659-2805

## Additional files

### Major datasets

The following datasets were generated:

| Author(s) | Year | Dataset title | Dataset URL | Database, license, and accessibility information |
|---|---|---|---|---|
| van Pee K, Neuhaus A, D'Imprima E, Mills DJ, Kuehlbrandt W, Yildiz O | 2016 | CryoEM structure of the membrane pore complex of Pneumolysin at 4.5A | https://www.ebi.ac.uk/pdbe/entry/pdb/5ly6 | Publicly available at the EMBL-EBI Protein Data Bank (accession no: 5LY6) |
| van Pee K, Neuhaus A, D'Imprima E, Mills DJ, Kuehlbrandt W, Yildiz O | 2016 | CryoEM structure of the membrane pore complex of Pneumolysin at 4.5A | http://www.ebi.ac.uk/pdbe/entry/emdb/EMD-4118 | Publicly available at the EMBL-EBI Protein Data Bank (accession no: EMD-4118) |
| van Pee K, Yildiz O | 2016 | Crystal structure of pneumolysin deletion mutant Delta146_147 | http://www.rcsb.org/pdb/explore/explore.do?structureId=5AOF | Publicly available at the RCSB Protein Data Bank (accession no: 5AOF) |
| van Pee K, Yildiz O | 2016 | Crystal structure of pneumolysin D168A mutant | http://www.rcsb.org/pdb/explore/explore.do?structureId=5AOE | Publicly available at the RCSB Protein Data Bank (accession no: 5AOE) |

The following previously published dataset was used:

| Author(s) | Year | Dataset title | Dataset URL | Database, license, and accessibility information |
|---|---|---|---|---|
| van Pee K, Yildiz O | 2016 | Crystal structure of wild type pneumolysin | http://www.rcsb.org/pdb/explore/explore.do?structureId=5aod | Publicly available at the RCSB Protein Data Bank (accession no: 5AOD) |

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
