## [Decision Letter]

Thank you for submitting your article "CryoEM structures of membrane pore and prepore complex reveal cytolytic mechanism of Pneumolysin" for consideration by *eLife*. Your article has been reviewed by three peer reviewers, one of whom, Sjors HW Scheres, is a member of our Board of Reviewing Editors, and the evaluation has been overseen by Richard Aldrich as the Senior Editor. The following individuals involved in review of your submission have agreed to reveal their identity: Christos G Savva (Reviewer #2).

The reviewers have discussed the reviews with one another and the Reviewing Editor has drafted this decision to help you prepare a revised submission.

Summary:

The study by van Pee and co-workers describes the cryo-EM structures of the cholesterol dependent cytolysin (CDC) pneumolysin (PLY) in the membrane-inserted and pre-pore states by single particle and tomography respectively. Although the first crystal structure of a CDC (perfringolysin O) in the water soluble state was solved almost 20 years ago, a high-resolution structure of the membrane-inserted form has been elusive due to the inherent structural heterogeneity of these complexes with the best results so far limited to the resolution of ~10-15Å where domains can be approximately identified but the exact mechanics of the transition from soluble to membrane-inserted cannot be seen. Despite the recent developments of direct electron detectors and better software that have pushed the resolution limits of Cryo-EM, this class of pore forming toxins still poses difficulties due to the irregular size and shape of the pores. In this study authors have succeeded in preparing more homogeneous samples by systematically screening various detergents. The end result is the first near-atomic resolution structure of a CDC in the membrane-inserted state, which provides valuable information for the mechanics of pore formation including the large movements of domains 2 and 3, which have not been resolved previously by other studies. In general the manuscript is well written and is an important step for the field of pore forming proteins of the CDC/MACPF family. Therefore, the three reviewers were overall positive about publishing this work, although several important issues were raised that should be addressed in a revised version.

Essential revisions:

1) The overall resolution of the single-particle reconstruction is limited to 4.5A, whereas the entire paper discusses the resulting atomic model as if this were a atomic-resolution structure. Locally the resolution is probably even worse than 4.5A, as for example the β strands are not well separated in Figure 1. Overall, too little information is given on the quality of the map and the corresponding atomic model. In fact, at 4.5A, the map alone will probably still leave considerable scope for registry or main chain tracing errors. Therefore, the authors should explicitly discuss these caveats where relevant in the main text. In addition, they should provide the following (supplementary?) information in their revision:

– A map that displays variations in local resolution (e.g. as calculated in resmap).

– A view of a segmented monomer from the sharpened map (possibly also colored by local resolution) would be helpful to further evaluate the quality of the map

– Zoomed-in views of density and model, or movie(s) of such, for various regions in the structure

– FSC curve between the refined model and the map. And because overfitting can be important at this resolution, preferably also FSC curves for refinement in one half-map, combined with FSC curves between the refined model and the other half-map.

– A more detailed description of the model building and refinement procedures, which is transparent about which parts of the model are less well defined than others.

2) At the reported resolution the salt bridge between Asp168 and Lys271 should be interpreted carefully. A close up of the EM densities around these residues should be provided. Although the authors have mutated Asp168 and seen a 20% reduction in hemolytic activity this is not conclusive of a salt-bridge disruption. Did mutation of Lys271 also result in the same phenotype? In addition the mutation seems to affect membrane binding as the authors mention (Figure 7 legend) and the ring attachment from the membranes (subsection “Determinants of lytic activity”). Does this imply that pores are formed but are then released? Previous AFM studies by the authors (van Pee et. Al, Nano Letters, 2016) suggest the process is irreversible. Some clarity and caution interpreting this salt bridge would improve the manuscript.

3) Also related to the limited resolution: interactions of the Trp-rich loop are discussed (Results section) but the electron density map in that region looks poorly resolved. This is a very important part of a CDC and the motif defines the family so another panel is required to show the quality of the electron density map in that region. (A close up stereo showing side-chains would help). There is great interest in seeing what conformation this loop adopts in the prepore and pore states since it is thought to control prepore to pore conversion. In Results paragraph two, the authors state that the undecapeptide is interacting with Thr405. This sentence should be re-worded to provide more detail. What is the nature of this interaction and again is the resolution of this area sufficient to justify this comment? From Figure 2 the loop itself does not seem to be resolved even at the Cα backbone level.

4) A major finding of this work is that domain 2 collapses but does not lose its structure (Results section). Earlier EM studies suggested it does lose its structure but modeling and FRET studies suggested otherwise. More discussion of this controversy is required. Importantly, the electron density of this key region should be shown in a figure to satisfy the reader that the structure has been maintained. (Figure 2 is not convincing enough. A stereo with side-chains focused on D2 only would be easier to evaluate). Some new detailed insights are discussed; namely, the remodeled helix-turn-helix motif and helix alpha3a. Can stereo figures of the electron density with side-chains displayed of these regions be shown?

5) The electron cryo-tomography of the prepore reveals a structure of 34 protomers in the prepore, a number expected based on numerous other reports. However, the CryoEM structure of the pore reveals a larger 42 protomer pore. This is unexpected but interestingly the authors note a range of ring structures were present with this being the most stable. So the question arises whether the pore structure is an artifact. Can the authors model a 34 protomer pore based on their 42 protomer structure and convince that the details they discuss in the 42 protomer pore are likely relevant to the smaller pore? Is the 20º incline in the β-hairpin maintained in modeling the smaller pore? Like the authors, other workers have found that PLY form linear oligomers in their crystals. The authors should discuss how linear oligomers fit into their detailed mechanism of pore formation.

6) The authors have skirted around another controversial issue of whether arcs and incomplete rings observed by other workers are physiological relevant as they are not observed in this work. Some discussion of why PLY does not form such structures in their hands but do in others would be insightful.

7) Subsection “Three stages of PLY pore formation”: In the previous AFM study (van Pee et. Al, Nano Letters, 2016) only 13% of the lower height rings were pre-pores on supported lipid membranes. This could be mentioned to indicate that the occurrence of a "late prepore" is rare event even on membranes without curvature.

8) The literature review, particularly in the Discussion, has been narrowly focused to a large degree on the PLY literature and should be broadened out to other CDCs, particularly to PFO where there is an extensive literature on how it forms pores. Does the current study support the major findings in the PFO literature and does it disagree with any aspects?

9) Parts of the Discussion should be moderated. The limited resolution and a few static snapshots do not reveal all the details of membrane insertion and pore formation. Such details will come gradually from a host of biochemical and biophysical approaches. The suggestion that the CryoEM structure forms a good basis for drug discovery to treat numerous Gram-positive infections is not supported by any results in the manuscript so should be restricted to a speculative claim about S. pneumoniae.

10) Please include crystallization and crystallography methods.

[Editors' note: further revisions were requested prior to acceptance, as described below.]

Thank you for resubmitting your work entitled "CryoEM structures of membrane pore and prepore complex reveal cytolytic mechanism of Pneumolysin" for further consideration at *eLife*. Your revised article has been favorably evaluated by Richard Aldrich (Senior editor) and three reviewers, one of whom is a member of our Board of Reviewing Editors.

The reviewers agreed that this manuscript has improved considerably and is almost ready for publication. However, there was also consensus that a few of the main points raised in the first round have not been dealt with satisfactorily. These are:

1) Over-interpretation of lower-resolution parts of the map in terms of the atomic model. Although the new supplementary figures now reveal the limitations of the map more clearly, over-interpretation in the text still remains a problem. Specifically, W433 is mentioned to move wrt the crystal structure, yet there is no density visible for W433 in Video 1, Figure 2—figure supplement 2 or 3 to support this statement. The map in this region is ~6Å. The same can be said for H407 that does also not have clear density. The authors should stay away from over-interpretation in this region and simply state that this domain is likely to contribute to the inter-molecular interactions that stabilize the pore, as they have done in the Discussion section, paragraph three. Likewise, there is no obvious density that connects D168 and K271 (Figure 5 instead shows connecting density between β-strands that is expected at resolutions lower than 4.7A). The resolution in this area is again ~6-7Å according to the local resolution estimation. The authors should be more cautious and can suggest that a salt bridge may exist as evident by the point mutant D168A, which has an 80% reduction in hemolytic activity.

2) The comment in the Discussion section: "The 4.5 Å cryoEM structure of the PLY pore complex thus paves the way towards the design of new drugs that inhibit toxin oligomerization and pore formation as a promising approach towards combating infections by wide-spread and dangerous Gram-positive pathogens" jars with the reply "the different CDCs seem to follow different pathways of pore formation…. Any more discussion of PFO papers would distract from the main focus of our work, which is on PLY." Therefore, the authors should focus the therapeutic claim focus to diseases that PLY has shown to contribute.

---

## [Author Response]

*Essential revisions:*

*1) The overall resolution of the single-particle reconstruction is limited to 4.5A, whereas the entire paper discusses the resulting atomic model as if this were a atomic-resolution structure.*

We are aware, and do not approve of the general tendency to overstate the resolution and quality of cryoEM maps. In this case, however, we have a high-resolution x-ray structure of the soluble form of PLY, and several of its domains move as rigid bodies. We would therefore be justified to describe at least these parts as an atomic resolution structure. Nevertheless we have refrained from such a description, and instead describe the structure in terms of an atomic model, which it no doubt is.

*Locally the resolution is probably even worse than 4.5A, as for example the β strands are not well separated in Figure 1. Overall, too little information is given on the quality of the map and the corresponding atomic model. In fact, at 4.5A, the map alone will probably still leave considerable scope for registry or main chain tracing errors.*

Register and chain-tracing errors in the β-strands of the barrel are actually not a problem, because the first and last stretches of each strand are fixed in the x-ray structure of domain 3. Together with the hydrophobic residues facing the lipid bilayer on the outside of the barrel and the polar and charged residues facing the inside, this leaves virtually no room for chain trace errors even at 4.5 Å. Moreover, we know that the β-strands in the barrel are well-resolved because of the sharp peak at 4.8 Å in the FSC curve (Figure 1—figure supplement 2), which is the characteristic distance between β-strands. We can therefore be sure that our atomic model of the β-barrel, which is the major new part of the structure, is correct and accurate within the constraints of the map.

*Therefore, the authors should explicitly discuss these caveats where relevant in the main text. In addition, they should provide the following (supplementary?) information in their revision:*

*– A map that displays variations in local resolution (e.g. as calculated in resmap).*

*– A view of a segmented monomer from the sharpened map (possibly also colored by local resolution) would be helpful to further evaluate the quality of the map*

It is true that the local resolution of some parts of the map, especially the loops connecting the long β-strands and one loop in the periphery of domain 4, is worse than 4.5 Å, but the resolution of domains 1, 2 and parts of domain 4 is significantly better. The local resolution estimate (new Figure 1—figure supplement 3) shows this clearly for the whole pore complex and for an individual protomer seen from different directions. The new supplementary figures demonstrate the excellent quality of the map, which resolves many of the tryptophan and tyrosine sidechains (Figure 2—figure supplement 1–Figure 2—figure supplement 3). This would be expected at 4 Å rather than at 4.5 Å resolution.

*– Zoomed-in views of density and model, or movie(s) of such, for various regions in the structure*

We included five new zoomed-in figures as supplements to Figure 2:

Figure 2—figure supplement 1: stereo diagram of the interface between domain 1 (red) and domain 3 (green) drawn as a ribbon-and-stick model. The local resolution estimate indicates 4 – 4.5 Å in this map region.

Figure 2—figure supplement 2: superposition of domain 4 of the membrane-inserted form (cryoEM structure, green) and the soluble form (x-ray structure, blue, pdb-id 5aod) in the cryoEM map drawn at two contour levels.

Figure 2—figure supplement 3: stereo diagrams of the undecapeptide (bright green) and domain 4 loops (blue) of the neighboring monomer on the membrane surface, seen from two directions.

Figure 2—figure supplement 4: stereo diagram of domain 3 (green) with the upper part of the β-barrel and the new helix-turn-helix motif (HTH) that forms the α-barrel inside the pore as a stick model in the cryoEM map. The local resolution estimate indicates 4 to 4.5 Å for the β-barrel outside the lipid bilayer and around 5 Å for the α-barrel. Also shown is the new helix α3a (red and yellow) of the neighboring monomer, indicating a potential ionic inter-monomer interaction of Lys196 with Asp59 that would stabilize the membrane-inserted pore complex.

Figure 2—figure supplement 5: stereo diagram of β-strands in domain 2 as stick model in the cryoEM map at two contour levels. The local resolution is around 4.5 Å.

*– FSC curve between the refined model and the map.*

The FSC curve between the model and the map is now included in Figure 1—figure supplement 2 (blue curve). Note that the model was used as built into the map, but not refined, for the reasons given below. Like the masked FSC, this curve shows a sharp peak at 4.8 Å, due to the strong contribution of the resolved long β hairpins in the barrel. The correlation factor at the steep drop beyond this peak at 4.5 Å is not 0.5 (which is an arbitrary value), but approximately 0.3. We draw the reviewers’ and editor’s attention to two papers, which will be familiar to the BRE of our study, that report similar thresholds for the map against model FSC of the 3.4 Å structure of the ribosome/Sec61 complex (Voorhees et al., Cell 2014) and the tri-snRNP spliceosome at 5.9 Å (Nguyen et al., Nature 2015).

*And because overfitting can be important at this resolution, preferably also FSC curves for refinement in one half-map, combined with FSC curves between the refined model and the other half-map.*

We can rule out overfitting for three reasons: (1) Substantial portions of the model were not built from scratch, but they were rigid-body fitted domains from our 2.4 Å x-ray structure of soluble PLY. These map regions cannot be overfitted. (2) The major new part of the membrane-inserted form of the PLY protomer is a β-barrel with 84 long β-hairpins of rigid, pre-defined geometry, so the risk of overfitting in this region is minimal. (3) The comparison of the phase-randomized vs masked FSC (Figure 1—figure supplement 2) indicates that there is no overfitted noise.

*– A more detailed description of the model building and refinement procedures, which is transparent about which parts of the model are less well defined than others.*

The individual domains of the PLY x-ray structure were fitted manually into the cryoEM map as rigid bodies with COOT. Where necessary, flexible or refolded protein regions were then refitted manually, followed by geometry regularization in COOT. We had described this procedure clearly in the methods of the original manuscript, but now describe it also in the main text, where it is less easily overlooked.

Fitting and refinement with Phenix or Refmac was tried but did not work, because these automatic procedures attempted to e.g. fit the membrane-inserted β-hairpins into the amphipol density around the hydrophobic part of the pore. The densities of amphipol and protein are comparable at 4-4.5 Å resolution, and only manual fitting can differentiate between them. Therefore, it was unfortunately not possible to employ automatic model fitting and refinement. It is well known that refinement of x-ray structures at similar resolutions also does not work, for the same reasons.

*2) At the reported resolution the salt bridge between Asp168 and Lys271 should be interpreted carefully. A close up of the EM densities around these residues should be provided.*

Even though this salt bridge is in the least well-defined map region near the tip of the long β-hairpins, there was significant density for it, as shown in the new panel G of Figure 5. We would like to point out that we did not look for this salt bridge, but deduced that there had to be one in this position from our chain trace, and then found the corresponding map density, which again shows there were no chain tracing errors in the β-barrel. Moreover, the interaction of adjacent β-hairpins that includes this salt bridge is in excellent agreement with the S-S crosslinking studies of Tweten and colleagues in the related PFO (Sato et al. 2013, NatChemBiol). This is now explained more fully in subsection “CryoEM structure of the PLY pore complex” paragraph four of the revised manuscript.

*Although the authors have mutated Asp168 and seen a 20% reduction in hemolytic activity this is not conclusive of a salt-bridge disruption. Did mutation of Lys271 also result in the same phenotype?*

The reviewer did not read our manuscript carefully. The mutation of Asp168 to alanine reduces the hemolytic activity by 80%, not to 80%! This is exactly the strong effect that one would expect from the disruption of a salt bridge stabilizing the membrane-inserted form. We do not see the point of mutating Lys271 as well.

*In addition the mutation seems to affect membrane binding as the authors mention (Figure 7 legend) and the ring attachment from the membranes (subsection “Determinants of lytic activity”). Does this imply that pores are formed but are then released? Previous AFM studies by the authors (van Pee et. Al, Nano Letters, 2016) suggest the process is irreversible. Some clarity and caution interpreting this salt bridge would improve the manuscript.*

We apologize if this was not clear in the original manuscript. In the original Figure 7 legend we wrote “Many PLY_D168A_ rings become detached from the liposomes, probably due to reduced binding affinity, and tend to break into fragments.” At this stage, the rings are of course prepores, and it is these, not the pores, that become detached and break into fragments. In the revised manuscript (subsection “Determinants of lytic activity” paragraph two) and Figure legend, we have clarified this point.

Note that the AFM study (van Pee et. al, Nano Letters, 2016, Figure 5) describes the transformation of membrane-attached rings into pores as irreversible, not the membrane attachment of the rings. Rings of this PLY mutant in the early prepore state detach from the membrane easily, but only before they (irreversibly) insert into the membrane to form the pore. This is now discussed in on paragraph two of the Discussion section.

*3) Also related to the limited resolution: interactions of the Trp-rich loop are discussed (Results section) but the electron density map in that region looks poorly resolved. This is a very important part of a CDC and the motif defines the family so another panel is required to show the quality of the electron density map in that region. (A close up stereo showing side-chains would help). There is great interest in seeing what conformation this loop adopts in the prepore and pore states since it is thought to control prepore to pore conversion. In Results paragraph two, the authors state that the undecapeptide is interacting with Thr405. This sentence should be re-worded to provide more detail. What is the nature of this interaction and again is the resolution of this area sufficient to justify this comment? From Figure 2 the loop itself does not seem to be resolved even at the Cα backbone level.*

The new Figure 2—figure supplement 2 shows domain 4 of PLY in the cryoEM (green) and x-ray structure from the inside of the ring. Differences between the undecapeptide structure in the soluble and pore forms of PLY are surprisingly small. The closest distance between Trp433 and Asp403 or Thr405 is 4-5 Å.

The stereo image of Figure 2—figure supplement 3 (see response to comment 1 above) shows the interactions of the D4 loops between two monomers. These interactions and their consequences are now discussed in more detail in the revised manuscript in paragraph three of the Discussion section.

*4) A major finding of this work is that domain 2 collapses but does not lose its structure (Results section). Earlier EM studies suggested it does lose its structure but modeling and FRET studies suggested otherwise. More discussion of this controversy is required. Importantly, the electron density of this key region should be shown in a figure to satisfy the reader that the structure has been maintained. (Figure 2 is not convincing enough. A stereo with side-chains focused on D2 only would be easier to evaluate). Some new detailed insights are discussed; namely, the remodeled helix-turn-helix motif and helix alpha3a. Can stereo figures of the electron density with side-chains displayed of these regions be shown?*

In the earlier cryoEM studies no intramolecular details were resolved and the rotation of domain 2 was interpreted as a “collapse”. To propose a rotation of domain 2 would have been more plausible but was not suggested. The FRET study (Ramachandran et al. 2005, PNAS) was interpreted in terms of a bending of domain 2 to account for the height change observed by AFM (Czajkowsky et al. 2004, PNAS). Our present cryoEM structure now shows that both interpretations were incorrect. We rewrote the corresponding text passage (Discussion section paragraph three) that now includes a reference to the FRET study. The AFM study had already been cited.

The new Figure 2—figure supplement 4 shows the remodeled helix-turn-helix (HTH) motif as stick models and Figure 2—figure supplement 5 the β-strands of domain 2.

*5) The electron cryo-tomography of the prepore reveals a structure of 34 protomers in the prepore, a number expected based on numerous other reports. However, the CryoEM structure of the pore reveals a larger 42 protomer pore. This is unexpected but interestingly the authors note a range of ring structures were present with this being the most stable. So the question arises whether the pore structure is an artifact.*

We do not think that the numbers of subunits we found in the PLY prepore and pore are at all unexpected. Tilley et al. (Cell, 2005) reported 31 ± 3 monomers in the PLY prepore, and two classes with 38 or 44 monomers in the PLY pore, in excellent overall agreement with our findings. Both the prepore and pore of Tilley et al. were in liposomes, so there can be no question that the number of subunits in our detergent-solubilized PLY pores is artefactual.

*Can the authors model a 34 protomer pore based on their 42 protomer structure and convince that the details they discuss in the 42 protomer pore are likely relevant to the smaller pore? Is the 20º incline in the β-hairpin maintained in modeling the smaller pore?*

It must have escaped the reviewer’s notice that we already built a 34-protomer model into the smaller pore of the subtomogram average, and that this was shown in Figure 6 of the original manuscript. No significant modification of the pore subunit, in particular not of the 20° incline of the β-hairpins relative to the membrane normal, was necessary to fit 34 copies of the pore-forming subunit into the 27 Å-resolution cryoET map. This is now pointed out more clearly in subsection “Three stages of PLY pore formation” of the revised manuscript

*Like the authors, other workers have found that PLY form linear oligomers in their crystals. The authors should discuss how linear oligomers fit into their detailed mechanism of pore formation.*

A description of how the linear arrays of the various x-ray structures relate to the prepore and pore form of PLY, and how this fits our detailed mechanism of pore formation can be found in our recent AFM paper (van Pee et al., Nano Lett 2016; supporting figure S2 and text relating to it).

*6) The authors have skirted around another controversial issue of whether arcs and incomplete rings observed by other workers are physiological relevant as they are not observed in this work. Some discussion of why PLY does not form such structures in their hands but do in others would be insightful.*

We certainly have no intention of skirting around this issue. Arcs and slits of PLY_wt_ are actually visible in Figure 7. This figure was there in the original manuscript, but we now point them out specially with coloured arrows. However, in contrast to LLO, PLY forms mostly complete rings.

*7) Subsection “Three stages of PLY pore formation”: In the previous AFM study (van Pee et. Al, Nano Letters, 2016) only 13% of the lower height rings were pre-pores on supported lipid membranes. This could be mentioned to indicate that the occurrence of a "late prepore" is rare event even on membranes without curvature.*

We thank the reviewer for this suggestion, which we have taken up in paragraph two of subsection “Three stages of PLY pore formation2” of the revised manuscript.

*8) The literature review, particularly in the Discussion, has been narrowly focused to a large degree on the PLY literature and should be broadened out to other CDCs, particularly to PFO where there is an extensive literature on how it forms pores. Does the current study support the major findings in the PFO literature and does it disagree with any aspects?*

Notwithstanding their high degree of sequence identity and similar monomer and oligomer structures, the different CDCs seem to follow different pathways of pore formation. Of the 21 papers on pore-forming toxins that we cite in our manuscript, more than half deal with PFO. Any more discussion of PFO papers would distract from the main focus of our work, which is on PLY.

*9) Parts of the Discussion should be moderated. The limited resolution and a few static snapshots do not reveal all the details of membrane insertion and pore formation. Such details will come gradually from a host of biochemical and biophysical approaches. The suggestion that the CryoEM structure forms a good basis for drug discovery to treat numerous Gram-positive infections is not supported by any results in the manuscript so should be restricted to a speculative claim about S. pneumoniae.*

The cryoEM structure of the PLY pore complex shows clearly the interactions of the secondary structure elements of neighboring protomers. The structure further shows how elements of secondary structure present in the soluble form refold and take on a completely different structure in the membrane. While the formation of the long membrane-inserted β-hairpins was predicted as such, no one could know or predict the exact structure of the β-strands and how they interact to form the membrane-perforating barrel. Other structural elements, including the inner α-barrel and the newly formed helix in domain 1, are entirely new. These elements are involved in inter- and intramolecular interactions in the pore complex. Even at 4.5 to 5 Å resolution there can be no reasonable doubt of their existence and interaction. Targeting these regions with newly developed drugs would be an elegant way to inhibit pore formation and infection, and thus an effective means of putting these pathogen out of action. We think that our discussion of this aspect of the PLY structure is reasonable and adequate.

10) Please include crystallization and crystallography methods.

Done

[Editors' note: further revisions were requested prior to acceptance, as described below.]

*The reviewers agreed that this manuscript has improved considerably and is almost ready for publication. However, there was also consensus that a few of the main points raised in the first round have not been dealt with satisfactorily. These are:*

*1) Over-interpretation of lower-resolution parts of the map in terms of the atomic model. Although the new supplementary figures now reveal the limitations of the map more clearly, over-interpretation in the text still remains a problem. Specifically, W433 is mentioned to move wrt the crystal structure, yet there is no density visible for W433 in Video-1, Figure 2—figure supplement 2 or 3 to support this statement. The map in this region is ~6Å. The same can be said for H407 that does also not have clear density. The authors should stay away from over-interpretation in this region and simply state that this domain is likely to contribute to the inter-molecular interactions that stabilize the pore, as they have done in the Discussion section, paragraph three.*

As explained in our first response letter, the location of most residues in domain 4, which hardly changes its conformation upon membrane insertion, is predefined by the crystal structure of the soluble PLY monomer. Although the resolution of the cryoEM map in the loop regions is lower, there is little room for misinterpreting the loop conformations. Nevertheless, we changed the manuscript text as requested by the reviewers.

The statement:

“In D4, Trp433 at the tip of the highly conserved undecapeptide loop (Figure 3) that renders PLY cholesterol-specific (Soltani et al., 2007b) was shifted by ~9 Å into the map density (Figure 2—figure supplement 2, Figure 2—figure supplement 3, Video 1). “

was changed to:

“In D4, the highly conserved undecapeptide loop (Figure 3) that renders PLY cholesterol-specific (Soltani et al., 2007b) was shifted by up to ~9 Å into the cryoEM map (Figure 2—figure supplement 2, Figure 2—figure supplement 3, Video 1).”

The statement:

“The short distances of Asp403 and Thr405 to Trp433 and of His407 to Try461 in adjacent monomers suggest a critical role of the D4 loop not only in receptor recognition, but also in oligomer formation.“

was changed to:

“The close proximity of loop β18/19 that contains Asp403, Thr405, and His407 of one monomer to loop β22/23 and the uncedapeptide containing Trp433 of the adjacent monomer suggests a critical role of these loops not only in receptor recognition, but also in oligomer formation.”

The statement:

“The conserved Trp433 in the undecapeptide loop seems to form a polar interaction with Asp403 and Thr405 in the loop that connects β18/19 in the neighbouring monomer, and Try461 interacts with His407 of the next monomer, most likely by π-stacking (Figure 2—figure supplement 2, Figure 2—figure supplement 3).”

was changed to:

“Loop β22/23 and the conserved undecapeptide loop with Trp433 at its tip appear to interact with loop β18/19 of the neighbouring monomer, thereby stabilizing the pore complex by inter- and intramolecular interactions (Figure 2—figure supplement 2, Figure 2—figure supplement 3).”

*Likewise, there is no obvious density that connects D168 and K271 (Figure 5 instead shows connecting density between β-strands that is expected at resolutions lower than 4.7A). The resolution in this area is again ~6-7Å according to the local resolution estimation. The authors should be more cautious and can suggest that a salt bridge may exist as evident by the point mutant D168A, which has an 80% reduction in hemolytic activity.*

The statement:

“[…]and (3) an ion bridge between Asp168 and Lys271 in adjacent b-strands of the pore barrel (Figure 5). Even though the hairpins at the ends of the trans-membrane b-strands are the least well-ordered part of the ring (Figure 1—figure supplement 3), there is significant map density for the salt bridge connecting them (Figure 5).”

was changed:

“[…]and (3) ionic interactions between charged sidechains in adjacent β-strands of the pore barrel (Figure 5). In particular, Asp168 and Glu170 in β–strand β7 are in a good position for forming a salt bridge with Lys271 in β–strand β10 of the next-door monomer (Figure 5).”

The statement:

“[…]we mutated Asp168 that is involved in ionic inter-subunit interactions[…]”

was changed to:

“[…]we mutated Asp168 that, as the structure suggests, might be involved in forming a salt bridge between adjacent subunits[…]”

*2) The comment in the Discussion section: "The 4.5 Å cryoEM structure of the PLY pore complex thus paves the way towards the design of new drugs that inhibit toxin oligomerization and pore formation as a promising approach towards combating infections by wide-spread and dangerous Gram-positive pathogens" jars with the reply "the different CDCs seem to follow different pathways of pore formation…. Any more discussion of PFO papers would distract from the main focus of our work, which is on PLY." Therefore, the authors should focus the therapeutic claim focus to diseases that PLY has shown to contribute.*

We apologize for not making this clear. As far as anyone knows, the mechanism of membrane insertion by rearrangement of the conserved domains 1 – 4 is the same for all CDCs. What we meant was that the exact pathways by which the membrane-inserted toxin monomers assemble into pores may differ between species. While most CDCs form circular pores, a small number of bacteria, for example *Listeria monocytogenes* or Streptococcus pyogenes, give rise to slit-like or arch-shaped pores. However, the structure of the membrane-inserted monomers in these slit-like or arch-shaped pores would be essentially the same as the membrane-inserted form of PLY. Therefore, compounds that interfere with refolding and membrane insertion of soluble monomers would prevent infection by Streptococcus pneumoniae and other CDC-producing Gram-positive bacteria. We have modified the last paragraph of the revised manuscript as follows:

“The mechanism of membrane insertion by rearrangement of the conserved domains 1 – 4 is likely to be the same for all CDCs. Therefore, compounds that interfere with refolding and membrane insertion of soluble monomers would prevent infection by Streptococcus pneumoniae and other CDC-producing Gram-positive bacteria that attack human cells with similar pore-forming toxins.

The 4.5 Å cryoEM structure of the PLY pore complex thus paves the way towards the design of new drugs that inhibit pore formation as a promising approach towards combatting infections by dangerous and wide-spread Gram-positive pathogens.”